# PIM kinases facilitate lentiviral evasion from SAMHD1 restriction via Vpx phosphorylation

Kei Miyakawa[1], Satoko Matsunaga[1], Masaru Yokoyama [2], Masako Nomaguchi[3], Yayoi Kimura[4], Mayuko Nishi[1], Hirokazu Kimura[5], Hironori Sato[2], Hisashi Hirano[4], Tomohiko Tamura[6], Hirofumi Akari[7,8], Tomoyuki Miura[8], Akio Adachi[3,9], Tatsuya Sawasaki[10], Naoki Yamamoto[11,12] & Akihide Ryo[1,4]

Lentiviruses have evolved to acquire an auxiliary protein Vpx to counteract the intrinsic host restriction factor SAMHD1. Although Vpx is phosphorylated, it remains unclear whether such phosphorylation indeed regulates its activity toward SAMHD1. Here we identify the PIM family of serine/threonine protein kinases as the factors responsible for the phosphorylation of Vpx and the promotion of Vpx-mediated SAMHD1 counteraction. Integrated proteomics and subsequent functional analysis reveal that PIM family kinases, PIM1 and PIM3, phosphorylate HIV-2 Vpx at Ser13 and stabilize the interaction of Vpx with SAMHD1 thereby promoting ubiquitin-mediated proteolysis of SAMHD1. Inhibition of the PIM kinases promotes the antiviral activity of SAMHD1, ultimately reducing viral replication. Our results highlight a new mode of virus–host cell interaction in which host PIM kinases facilitate promotion of viral infectivity by counteracting the host antiviral system, and suggest a novel therapeutic strategy involving restoration of SAMHD1-mediated antiviral response.

[1] Department of Microbiology, Yokohama City University School of Medicine, Kanagawa 236-0004, Japan. [2] Pathogen Genomics Center, National Institute of Infectious Diseases, Musashi Murayama, Tokyo 208-0011, Japan. [3] Department of Microbiology, Tokushima University Graduate School of Medical Science, Tokushima 770-8503, Japan. [4] Advanced Medical Research Center, Yokohama City University, Kanagawa 236-0004, Japan. [5] School of Medical Technology, Faculty of Health Sciences, Gunma Paz University, Gunma 370-0006, Japan. [6] Department of Immunology, Yokohama City University School of Medicine, Kanagawa 236-0004, Japan. [7] Laboratory of Infectious Disease Model, Institute for Frontier Life and Medical Sciences, Kyoto University, Kyoto 606-8507, Japan. [8] Center for Human Evolution Modeling Research, Primate Research Institute, Kyoto University, Aichi 484-8506, Japan. [9] Department of Microbiology, Kansai Medical University, Osaka 573-1010, Japan. [10] Proteo-Science Center, Ehime University, Ehime 790-8577, Japan. [11] National Institute of Infectious Diseases, Tokyo 162-8640, Japan. [12] Tokyo Medical and Dental University, Tokyo 113-8519, Japan. Correspondence and requests for materials should be addressed to A.R. (email: aryo@yokohama-cu.ac.jp)

Reciprocal interplay between virus and host proteins plays important roles in both promoting and suppressing viral replication[1,2]. Several host proteins exert significant antiviral effects at multiple steps of the viral life cycle, but viruses have evolved to counteract such host defenses and achieve efficient replication[3]. Moreover, by hijacking host cell machinery, viruses take advantage of cellular regulatory systems to promote successful progeny production and spread.

Sterile alpha motif and histidine-aspartate domain-containing protein 1 (SAMHD1) was identified as an inhibitor of several lentiviruses, including human immunodeficiency virus (HIV). SAMHD1 acts as a deoxynucleotide triphosphate (dNTP) phosphohydrolase[4], and thus lowers the concentration of dNTPs required for viral reverse transcription in infected non-dividing myeloid cells and resting T cells[5,6]. On the other hand, viruses can overcome SAMHD1-mediated viral restriction through utilizing viral protein X (Vpx), an accessory protein encoded by human immunodeficiency virus type 2 (HIV-2) and some strains of simian immunodeficiency virus (SIV)[7,8]. In fact, Vpx hijacks a Cullin-RING ubiquitin ligase (CRL) complex that associates with the host E3 ubiquitin ligase components, i.e., DCAF1, DDB1, and CUL4, to promote the ubiquitin-mediated proteolysis of SAMHD1[9]. These events, by suppressing the host innate immune system, ultimately result in efficient viral replication.

Host protein kinases directly phosphorylate viral proteins during infection and control the efficiency of viral replication. In HIV infection, the human protein kinases ERK2[10] and atypical PKC[11] phosphorylate the HIV-1 Gag protein to regulate viral assembly, release, and infectivity. Moreover, phosphorylation of the HIV-1 capsid by maternal embryonic leucine zipper kinase (MELK) promotes viral uncoating and cDNA synthesis[12]. Previous reports demonstrated that Vpx is also phosphorylated during infection[13–15]. Although phosphorylation of Vpx seems to influence its nuclear import or packaging into virions[14], it remains uncertain whether Vpx phosphorylation contributes to its crucial function in the degradation of SAMHD1. Moreover, it is not clear which host kinases are responsible for the functional phosphorylation of Vpx.

In this study, we perform a comprehensive proteomic analysis for the molecular interactions between human protein kinases and HIV-2 Vpx, and identify PIM kinases as regulatory factors for Vpx-mediated SAMHD1 degradation. Our findings thus reveal a regulatory mechanism of virus–host interaction that governs viral escape from an intrinsic cellular immune defense via the post-translational modification of viral protein.

## Results

**Identification of host kinases that phosphorylate Vpx**. To comprehensively survey host kinase(s) that might bind to HIV-2 Vpx and influence its function, we performed an in vitro protein–protein interaction screen using the amplified luminescent proximity homogenous assay (AlphaScreen). Full-length Vpx and 412 human protein kinases were separately synthesized using the wheat cell-free protein production system, and each kinase was screened for interaction with Vpx (Supplementary Fig. 1). When a relative light unit per cutoff (RLU/Co) ratio of ≥1.5 was set as the threshold, 50 host kinases were identified as potentially interacting with Vpx in vitro (Fig. 1a). To search for biologically significant kinases, we next performed a cell-based interaction analysis using the recently developed NanoLuc bioluminescence resonance energy transfer (NanoBRET) assay (Supplementary Fig. 2). Among the 50 candidates selected in the first screening, we identified three host kinases, namely, PIM1, PIM3, and PDK1 (Fig. 1b), that interacted intimately with Vpx in living cells, as revealed by the fact that these kinases yielded the

highest BRET signal (RLU/Co ratio of ≥20). Since two of these proteins, PIM1 and PIM3, are members of PIM (Proviral Integration site for Moloney murine leukemia virus) family of kinases, we decided to focus on PIM kinase family. An immunoprecipitation analysis further revealed that PIM1 and PIM3, but not another PIM kinase family member, PIM2, stably interacted with Vpx (Fig. 1c). To functionally analyze the PIM family kinases in relation to Vpx, we investigated whether the PIM kinases could directly phosphorylate Vpx in vitro. To this end, biotinylated Vpx was incubated with recombinant kinases, and this mixture was then processed for an in vitro kinase assay using radioactive adenosine 5′-triphosphates. The result revealed that phosphorylation of Vpx could be performed by PIM3, and to a lesser extent by PIM1, while PIM2 phosphorylated much less Vpx protein than PIM1 and PIM3 in vitro (Fig. 1d). We also detected Vpx phosphorylation by FYN kinase, as reported previously[15] (Fig. 1d). Together, these results indicate that the PIM family kinases, namely, PIM1 and PIM3, are previously uncharacterized host mediators of Vpx phosphorylation.

**PIM kinases phosphorylate the Ser13 residue of Vpx**. To determine the sites on Vpx phosphorylated by PIM kinases, we performed a proteomic analysis. For these experiments, cells were transfected with Vpx with or without PIM3, and immunoprecipitated Vpx was subjected to liquid chromatography tandem-mass spectrometry analysis. Subsequent in-depth data analysis revealed that PIM3 could phosphorylate three amino acid residues (Ser13, Thr88, and Ser101) of Vpx (Fig. 2a, b and Supplementary Fig. 3). Multiple sequence alignment revealed that the amino acid sequence N-terminal to the Ser13 phospho-acceptor site is highly conserved (98.7%) among HIV-2 and SIV isolates (SIVmac and SIVsmm) (Fig. 2c). To further investigate the roles of PIM kinases in Vpx function, we created a phospho-specific antibody that specifically recognizes site-specific phosphorylation of Vpx Ser13. Using this antibody, we confirmed that PIM1 and PIM3, but not PIM2, could phosphorylate Vpx Ser13 in vitro (Fig. 2d). Ectopic expression of wild-type PIM3, but not its kinase-dead mutant (K69M)[16], significantly enhanced the phosphorylation of Vpx Ser13 in cells (Fig. 2e). We confirmed that this phosphorylation was detected in wild-type Vpx, but not its S13A mutant (Supplementary Fig. 4a). Moreover, we found that another candidate, PDK1, selected by initial screening, failed to phosphorylate this site of Vpx (Supplementary Fig. 4b). Interestingly, PIM3 could phosphorylate SIVmac Vpx (Fig. 2f), suggesting that the PIM kinase-mediated Vpx phosphorylation at Ser13 is a conserved post-translational modification across the HIV-2 and SIV lineages.

**Low infectivity of HIV-2 bearing Vpx S13A**. To investigate the functional impact of Vpx Ser13 phosphorylation, we next performed a single-cycle viral infection assay using HIV-2 encoding a luciferase reporter gene (HIV-2-Luc) (Fig. 3a). In this assay, we used wild-type HIV-2-Luc and viruses encoding either Vpx-S13A or Vpx-Q76A (lacking the ability to interact with the CRL4 E3 complex[7,17]); the Vpx-null (ΔVpx) derivative was used as a control. The amounts of viral production and Vpx incorporation into virions were almost equivalent in all of these viruses (Supplementary Fig. 5). Notably, the infectivity of Vpx-S13A virus was significantly lower than that of the wild-type virus in SAMHD1-positive (Monomac6-derived) macrophages (Fig. 3b). However, this was not the case in SAMHD1-negative (U937-derived) macrophages (Fig. 3c). Immunoblot analysis revealed that the levels of SAMHD1 were reduced in macrophages infected with wild-type HIV-2, but this reduction was less prominent in cells infected with the Vpx-S13A virus

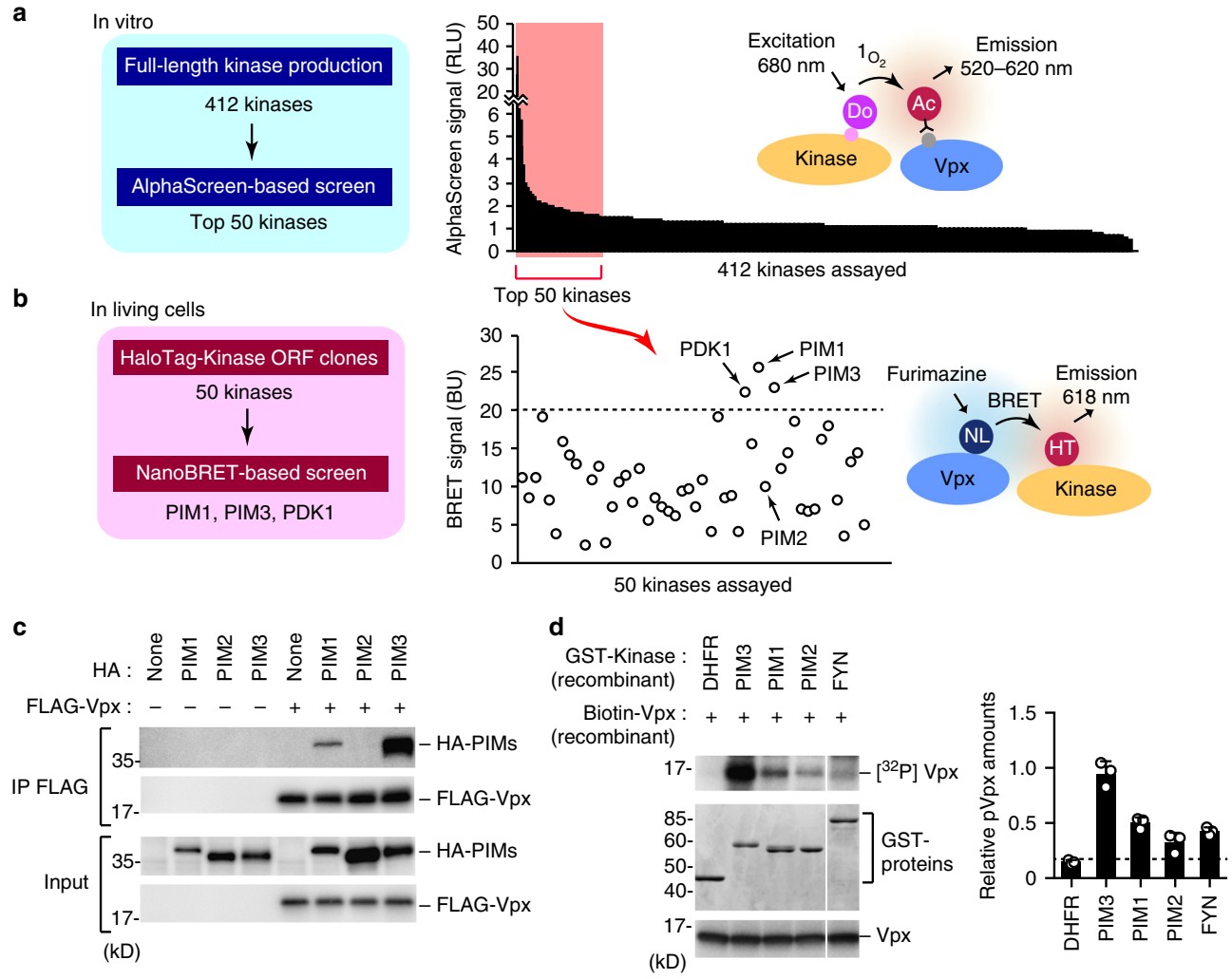

**Fig. 1** Identification of protein kinases responsible for Vpx phosphorylation. **a** Schematic representation of the AlphaScreen-based luminescent system to screen for human kinases that interact with HIV-2 Vpx in vitro. Streptavidin-coated donor beads (Do) and anti-FLAG antibody conjugated to protein A acceptor beads (Ac) were mixed with recombinant Vpx and kinase proteins, respectively, and AlphaScreen signals were detected if the two proteins were close each other (See Supplementary Fig. 1 for detail procedure). When a relative light unit per cutoff (RLU/Co) ratio of ≥1.5 was set as the threshold, 50 kinases were identified as potentially Vpx-interacting proteins. The AlphaScreen was performed in duplicate for each sample. **b** NanoBRET-based secondary screen to identify Vpx-interacting proteins in living cells. HEK293 cells were co-transfected with NanoLuc (NL)-conjugated Vpx and HaloTag (HT)-fused kinase expression vectors, and then HT-618 ligand and furimazine substrate were added to cells (see Supplementary Fig. 2 for detail procedure). When a relative BRET signal of ≥20 was set as the threshold, three kinases were identified as Vpx-interacting proteins in living cells. The NanoBRET was performed in duplicate for each sample. **c** Immunoprecipitation analysis for PIM family kinases to interact with Vpx. HEK293 cells were transfected with HA-tagged PIM kinases and FLAG-Vpx. Forty-eight hours after transfection, cells were lysed and then immunoprecipitated with anti-FLAG antibody followed by immunoblot analysis. **d** Biochemical approach for detecting the phosphorylation of Vpx by PIM family kinases. Recombinant biotinylated Vpx and indicated kinase proteins (GST-tagged) were produced in a wheat-germ cell-free system, and then mixed and incubated with $^{32}$P-labeled adenosine 5′-triphosphate. Mixtures were subjected to autoradiography (top) or immunoblotting with anti-GST antibody (middle) or anti-Vpx (bottom). Dihydrofolate reductase (DHFR) was used as a negative control. Bar graph indicates the percentage of phospho-Vpx in total Vpx, as determined by densitometry ($n = 3$, mean ± s.d.). Source data are provided as a Source Data file

(Fig. 3b). We also found that Vpx-S13A virus exhibited much lower infectivity than wild-type virus with sustained expression of SAMHD1, in primary human monocyte–derived macrophages (Fig. 3d). We performed parallel experiments using a virus carrying phosphomimetic Vpx-S13E. Both infectivity and SAMHD1 degradation of the Vpx-S13E virus were comparable to those of the wild-type virus (Supplementary Fig. 6), demonstrating the functional significance of Ser13 phosphorylation. We next performed a multi-cycle HIV-2 replication analysis using SAMHD1-positive (THP1-derived) macrophages (Fig. 3e). Consistent with previous reports[18], HIV-2 virus bearing Vpx-Q76A or ΔVpx exhibited significantly lower

replication competency than the wild-type virus in THP1-derived macrophages (Fig. 3f). In line with the results of single-cycle infection, HIV-2 carrying Vpx-S13A had lower replication capacity (Fig. 3f), presumably due to the persistent SAMHD1 expression (Fig. 3g). These results were also confirmed by a flow cytometry analysis in which infected cells were gated (Supplementary Fig. 7a). Our results revealed that S13A virus exhibited lower viral infectivity and less potently decreased SAMHD1 levels than the wild-type virus (Supplementary Fig. 7b, c). Together, these data indicate that the Ser13 residue of Vpx is functionally significant in SAMHD1 degradation and lentiviral replication in human macrophages.

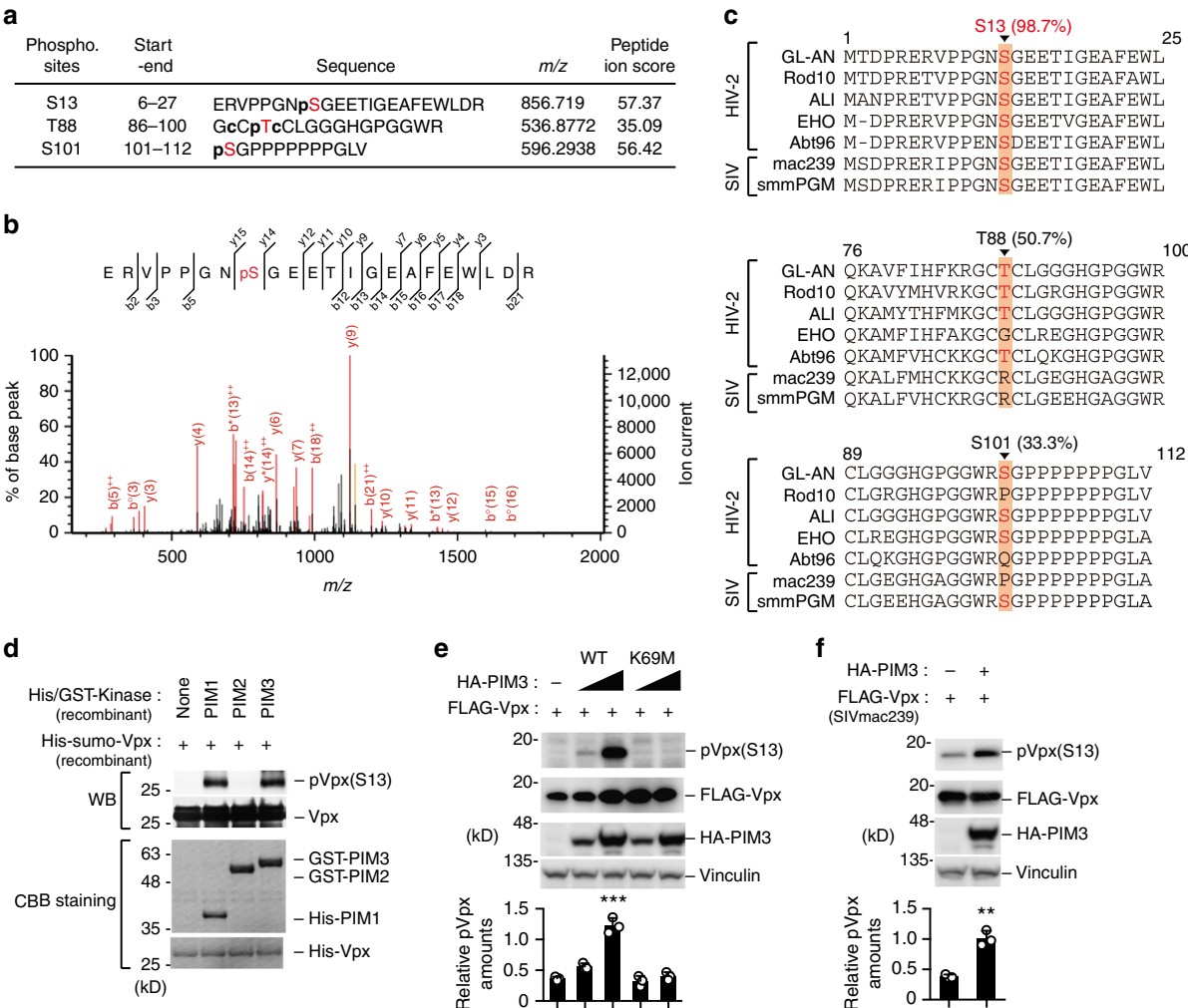

**Fig. 2** PIM kinases directly phosphorylate the Ser13 residue of Vpx. **a**, **b** Proteomic analysis revealed potential phospho-acceptor sites within Vpx by PIM3. HEK293 cells were transfected with plasmid vector encoding Vpx together with either empty vector or PIM3. Immunoprecipitated Vpx was subjected to liquid chromatography tandem-mass spectrometry. Phosphopeptides of Vpx proteins detected by proteomic analysis in PIM3-expressing HEK293 cells are described (**a**). p and c in a sequence indicates phosphorylation and carbamidomethylation, respectively. Typical spectra of S13 phosphopeptides are shown in **b**. Representative data of two experiments are shown. **c** Amino acid sequences around the potential phospho-acceptor sites were alignment of indicated Vpx of HIV-2/SIV strains. Note that the Ser13 phospho-acceptor site is highly conserved (98.7%). **d** Detection of in vitro phosphorylation of Vpx by PIM kinases with phospho-specific antibody. Recombinant Vpx and indicated kinase proteins (His-tagged for PIM1 and GST-tagged for PIM2 and PIM3) were incubated and subjected to immunoblotting (top) or CBB staining (bottom). **e** PIM3, but not its kinase-defective mutant, increases Vpx Ser13 phosphorylation. HEK293 cells were co-transfected with expression vectors encoding HA-PIM3 (WT or its kinase-dead mutant K69M) and FLAG-Vpx derived from HIV-2$_{GL-AN}$. Forty-eight hours after transfection, cells were harvested and subjected to immunoblot analysis. Bar chart indicates the percentage of phospho-Vpx in total Vpx as determined by densitometry of immunoblots. **f** PIM3 phosphorylates SIV-Vpx. HEK293 cells were transfected with Vpx derived from SIVmac239 and subjected to immunoblot analysis. Bar chart indicates the percentage of phospho-Vpx in total Vpx. All graphs are presented as a mean ± s.d. ($n = 3$). **$P < 0.01$; ***$P < 0.001$, two-tailed unpaired $t$-test. Source data are provided as a Source Data file

**Vpx Ser13 phosphorylation promotes Vpx–SAMHD1 interaction**. The Ser13 residue of Vpx was located at the vicinity of the SAMHD1-binding site, as well as in part of the DCAF1-binding element (Fig. 4a). To further delineate the molecular mechanism underlying Ser13 phosphorylation, we performed a molecular dynamics simulation based on the previously solved structure of the Vpx–DCAF1–SAMHD1 complex[19] to predict phosphorylation-induced conformational changes. The models constructed in this study included either unphosphorylated or phosphorylated Vpx, and revealed that phosphorylation status does not cause an obvious difference in the structure of Vpx (Fig. 4b). However, our molecular simulation suggested that the phosphorylation of Vpx Ser13 may create an additional hydrogen bond with SAMHD1 Ser616, leading to stabilization of the Vpx–SAMHD1 interaction (Fig. 4c).

To confirm the results of the molecular dynamics simulation, we investigated the interaction between Vpx and SAMHD1 in living cells using NanoBRET. We observed a constantly high BRET signal, reflecting Vpx–SAMHD1 interaction, in the case of wild-type Vpx, whereas the signal was significantly reduced in the Vpx-S13A mutant (Fig. 4d). On the other hand, the Vpx-S13A mutation had no prominent effect on the interaction between Vpx and DCAF1 (Fig. 4e). These trends were also observed in an immunoprecipitation analysis in which the Vpx-S13A mutant bound SAMHD1 to a lesser extent than wild-type Vpx (Fig. 4f±h).

Moreover, Vpx-S13A was less capable of poly-ubiquitinating SAMHD1 than wild-type Vpx (Fig. 4i). Consistent with this, we found that Vpx-S13A was less able than wild-type Vpx to degrade SAMHD1, despite similar expression levels (Supplementary

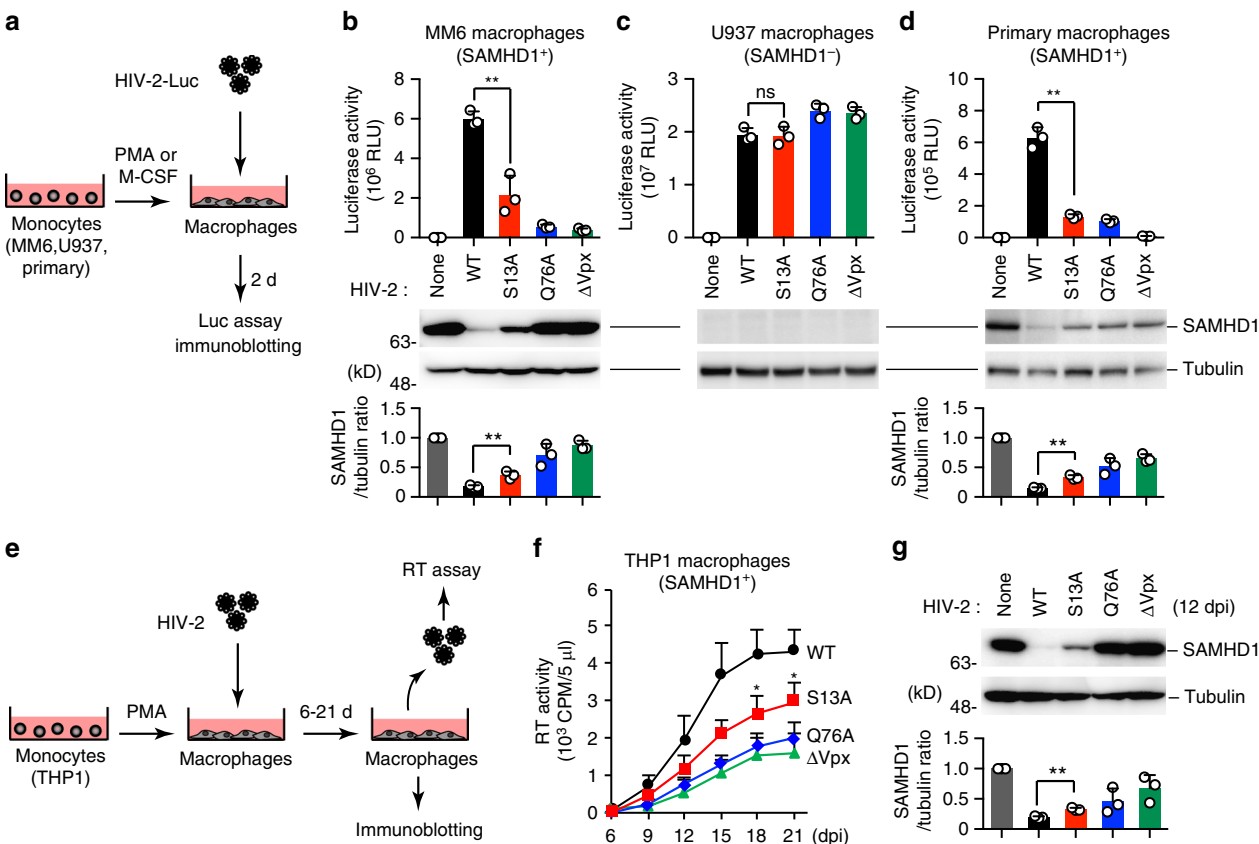

**Fig. 3** Low infectivity of HIV-2 bearing Vpx-S13A in myeloid cells. **a–d** Single-cycle HIV-2 infection assays in various human macrophages. Schematic representation of the experimental design (**a**). Briefly, monocytic cells were differentiated into macrophages by PMA or M-CSF and then infected with the indicated HIV-2 encoding luciferase (Luc) reporter gene. Forty-eight hours after infection, intracellular luciferase activity was measured for Monomac6 (MM6) (**b**), U937 (**c**) and primary monocyte-derived macrophage (**d**). Endogenous SAMHD1 expression in infected cells is also shown. Bar charts below the blot indicate the ratio of SAMHD1 over Tubulin, as determined by densitometry. **e**, **f** Multi-cycle HIV-2 replication kinetics in THP1-derived macrophages. Schematic representation of the experimental procedure (**e**). Cells were infected with wild-type HIV-2$_{GL-AN}$ and its Vpx-S13A, Q76A, and Vpx-deficient mutants. RT activity of each supernatant was measured at the indicated time points (**f**). **g** SAMHD1 expression in the indicated HIV-2–infected THP1-macrophages, 12 days after infection. Bar chart below the blot indicates the ratio of SAMHD1 over Tubulin, as determined by densitometry. All graphs are presented as a mean ± s.d. ($n = 3$). *$P < 0.05$; **$P < 0.01$; ns, not significant, two-tailed unpaired $t$-test. Source data are provided as a Source Data file

Fig. 8a). In addition, we performed a cycloheximide assay to examine the protein stability of WT and the S13A mutant; the results revealed that both Vpx proteins had similar protein half-lives (Supplementary Fig. 8b), indicating that the Ser13 phosphorylation affects its functional activity rather than its stability. Interestingly, our analysis revealed that SAMHD1-S616A mutant was resistant to the Vpx-mediated proteolysis (Supplementary Fig. 9), indicating that Ser616 of SAMHD1 was indeed responsible for the interaction with Vpx. Together, these results suggest that phosphorylation of Vpx Ser13 stabilizes the Vpx–SAMHD1 interaction, thereby promoting ubiquitin-mediated proteolysis of SAMHD1.

**PIM kinase inhibition prevents SAMHD1 degradation by Vpx.** The results described above indicate that PIM kinase–mediated phosphorylation of lentiviral Vpx regulates its function in down-regulating SAMHD1. We next examined the effect of PIM kinase inhibition on lentiviral replication. To this end, Monomac6 macrophage were stably transduced with shRNAs targeting PIM1 and PIM3, and then infected with HIV-2-Luc (Fig. 5a). Simultaneous knockdown of PIM1 and PIM3 significantly blocked viral infection through the prevention of SAMHD1 degradation (Fig. 5b, c), whereas depletion of either PDK1 or PIM2 had no effect on viral infectivity (Supplementary Fig. 10). Notably, the

infectivity of the Vpx-S13A virus was comparable to that of the wild-type virus, albeit at a relatively low level, in PIM1/3-depleted macrophages (Fig. 5b, c). Moreover, when PIM1 and PIM3 kinases were depleted in producer cells, viral infectivity was not significantly affected (Fig. 5d–f). These results together indicate that PIM1 and PIM3 have a profound effect upon Vpx Ser13 in target cells, but not in producer cells, implying their action on the early stages of viral life cycle.

Currently, several potential pharmacological inhibitors of PIM kinase are under development, and a few are in clinical trials in anti-cancer treatments. To determine whether PIM kinase inhibitors could block HIV-2 infection, we treated primary human monocyte–derived macrophages with the pan-PIM kinase inhibitor AZD1208[20] prior to HIV-2-Luc infection (Fig. 6a). As expected, this compound inhibited HIV-2 infection in a dose-dependent manner while preventing SAMHD1 degradation (Fig. 6b, c). However, this effect was not observed in cells infected with the Vpx-S13A virus, which has low infectivity in nature (Fig. 6b, c). Moreover, we found that the inhibitory effects of AZD1208 were significantly attenuated when SAMHD1 was depleted in target macrophages (Fig. 6d, e). AZD1208 was not cytotoxic at effective concentrations (~1 μM), but interestingly, this drug exhibited a greater cytotoxic effect in Monomac6-derived cells than in primary macrophages (Supplementary

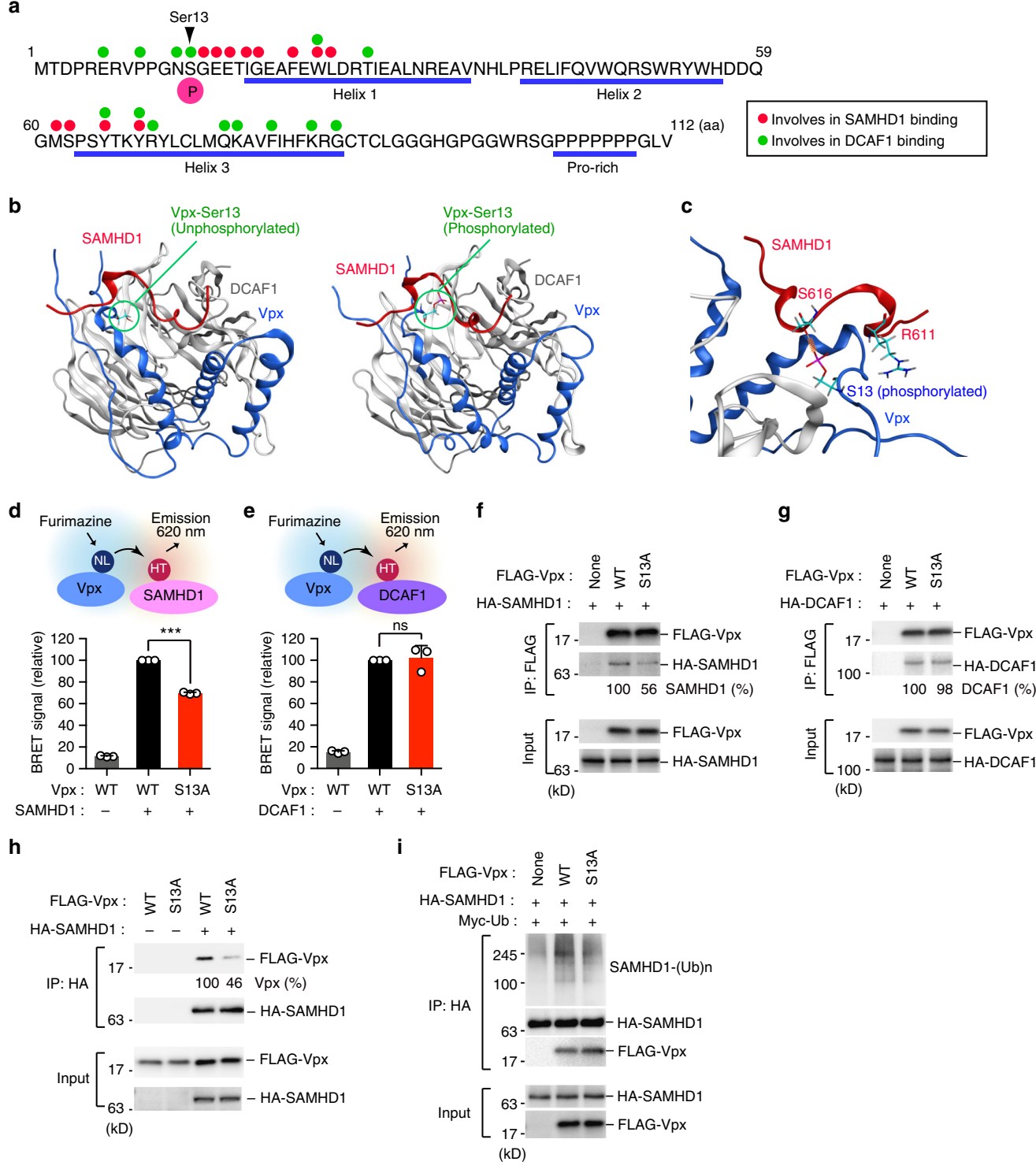

Fig. 11). Another PIM inhibitor, SGI1776[21], also exhibited inhibitory effects on HIV-2 infection similar to those of AZD1208 (Fig. 6f). Together, our data clearly indicate that selective PIM kinase inhibitors potentiate the antiviral activity of SAMHD1 by suppressing Vpx phosphorylation and function.

**PIM kinase inhibitors block lentiviral replication**. To clarify the effect of PIM kinase inhibitors on multi-cycle viral replication in primary cells, we treated human primary macrophages with HIV-2 in the presence or absence of PIM kinase inhibitors for 7 days,

and then calculated the yield of progeny virus (Fig. 7a). Notably, our results demonstrated that treatment with only 1 μM of either AZD1208 or SGI1776 could block the HIV-2 replication (Fig. 7b). Notably, these compounds exerted no observable effect on the replication of HIV-1 that does not encode Vpx (Fig. 7b).

The results described above indicated that the Ser13 residue of Vpx is highly conserved within the HIV-2/SIVmac/SIVsmm lineage (Fig. 2c); consistent with this, SIVmac Vpx was also phosphorylated by PIM3 (Fig. 2f). Accordingly, we investigated whether PIM kinase inhibitors could block SIVmac replication

**Fig. 4** Vpx Ser13 phosphorylation strengthens the Vpx–SAMHD1 interaction. **a** Primary structure of Vpx, showing the SAMHD1-binding (red) and DCAF1-binding (green) motifs[19]. Arrow indicates the residue of phospho-Ser13. **b** Molecular dynamics simulations of Vpx (blue) coupled with peptides derived from SAMHD1 (red) and DCAF1 (gray) in the absence (left) or presence (right) of Vpx Ser13 phosphorylation. **c** Expanded view of a predicted hydrogen bond between phospho-Vpx and SAMHD1 (shown in an orange line). **d**, **e** NanoBRET assays of HEK293 cells expressing NanoLuc-conjugated Vpx and either SAMHD1 (**d**) or DCAF1 (**e**) fused with HaloTag. Forty-eight hours after transfection, intracellular NanoBRET signals were calculated. All graphs are presented as a mean ± s.d. ($n = 3$). ***$P < 0.001$; ns, not significant, two-tailed unpaired $t$-test. **f–h** Vpx-S13A binds SAMHD1 to a lesser extent than wild-type Vpx. HEK293 cells expressing HA-SAMHD1 (**f**) or HA-DCAF1 (**g**) and either FLAG-Vpx WT or S13A were lysed; and the resultant lysates were immunoprecipitated with anti-FLAG (**f**, **g**) or anti-HA antibodies (**h**); the precipitants were analyzed by immunoblotting. The numerical values below the blots show the signal intensities of the indicated precipitants, as determined by densitometry. **i** Vpx-S13A is less capable of poly-ubiquitinating SAMHD1 than wild-type Vpx. HEK293 cells were cotransfected with plasmids encoding HA-SAMHD1, FLAG-Vpx (WT or S13A), and Myc-ubiquitin. The cells were then treated with MG132 for 18 h and harvested. Cell lysates were immunoprecipitated with anti-HA antibody, and the Vpx-induced ubiquitination of SAMHD1 was detected by immunoblotting with anti-Myc antibody. Source data are provided as a Source Data file

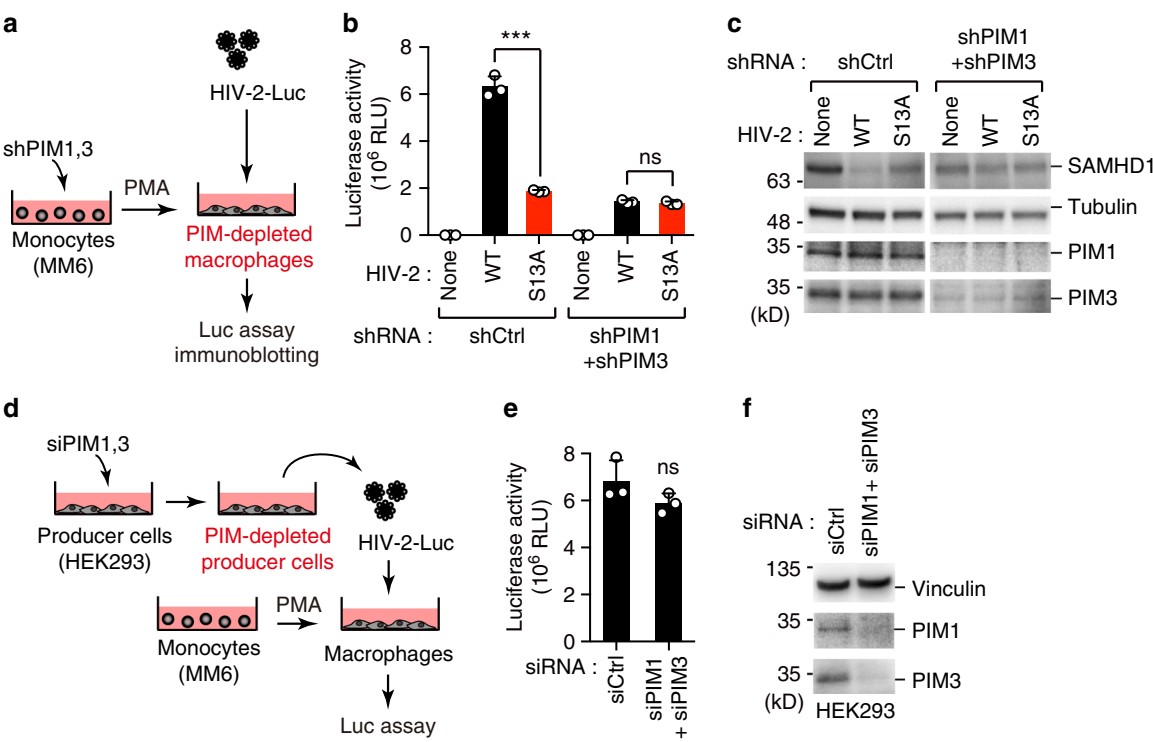

**Fig. 5** Depletion of PIM kinases attenuates anti-SAMHD1 activity of Vpx. **a–c** Knockdown of PIM1 and PIM3 results in reduced viral replication. Schematic representation of the experimental system (**a**). Monomac6 (MM6) cells were stably transduced with shRNAs targeting PIM1 and PIM3. Cells were then differentiated into macrophages and infected with HIV-2 luciferase (Luc) reporter virus. Forty-eight hours later, the cells were harvested and subjected to luciferase assay (**b**) and immunoblotting (**c**) to measure HIV-2 infectivity and SAMHD1 expression, respectively. **d–f** Knockdown of PIM1 and PIM3 in producer cells does not affect late stage of virus replication cycle. Schematic representation of the experimental procedure (**d**). HEK293 cells were treated with either control or PIM-targeted siRNA for 24 h before transfection with an HIV-2 molecular clone. After 48 h, virus-containing culture supernatants were harvested. Equal amounts of HIV-2 were then used to infect Monomac6-derived macrophages. Forty-eight hours later, cells were subjected to luciferase assay to measure HIV-2 infectivity (**e**). Expressions of PIM1 and PIM3 in HEK293 producer cells are also shown (**f**). All graphs are presented as a mean ± s.d. ($n = 3$). ***$P < 0.001$; ns, not significant, two-tailed unpaired $t$-test. Source data are provided as a Source Data file

ex vivo. To this end, we prepared virally infected mononuclear cell mixtures from the lymph nodes of SIVmac-infected rhesus macaques. Prior to brief stimulation, only macrophages adhered to the bottoms of dishes were recovered. The cells were then treated with PIM kinase inhibitors for 7 days (Fig. 7c). Notably, treatment with either AZD1208 or SGI1776 could significantly block viral replication (Fig. 7d). Taken together, these results confirm that PIM kinases are potent therapeutic targets for lentiviral infection.

## Discussion

In recent years, a great deal of effort has been made to discover a new mode of virus–host protein interactions to understand the molecular basis underlying virus infection[1,2]. During viral

replication, viruses take advantage of cellular enzymes to optimize viral protein function in infected cells. Phosphorylation of viral proteins by host kinases is critical for efficient viral replication and pathogenesis[22]. Several viruses encode their own protein kinases for these modifications (e.g., herpes simplex virus-thymidine kinase), but lentiviruses have no such virally encoded kinase. Instead, these viruses have evolved to hijack host protein kinases for viral protein phosphorylation in order to ensure efficient viral replication. Indeed, a previous study predicted a large number of as-yet-unidentified phosphorylation sites on viral proteins[23]. Identification of host kinases involved in viral protein phosphorylation and their functional modification could facilitate the development of new antiviral strategies[24]. In our current study, by screening for human protein kinases that target

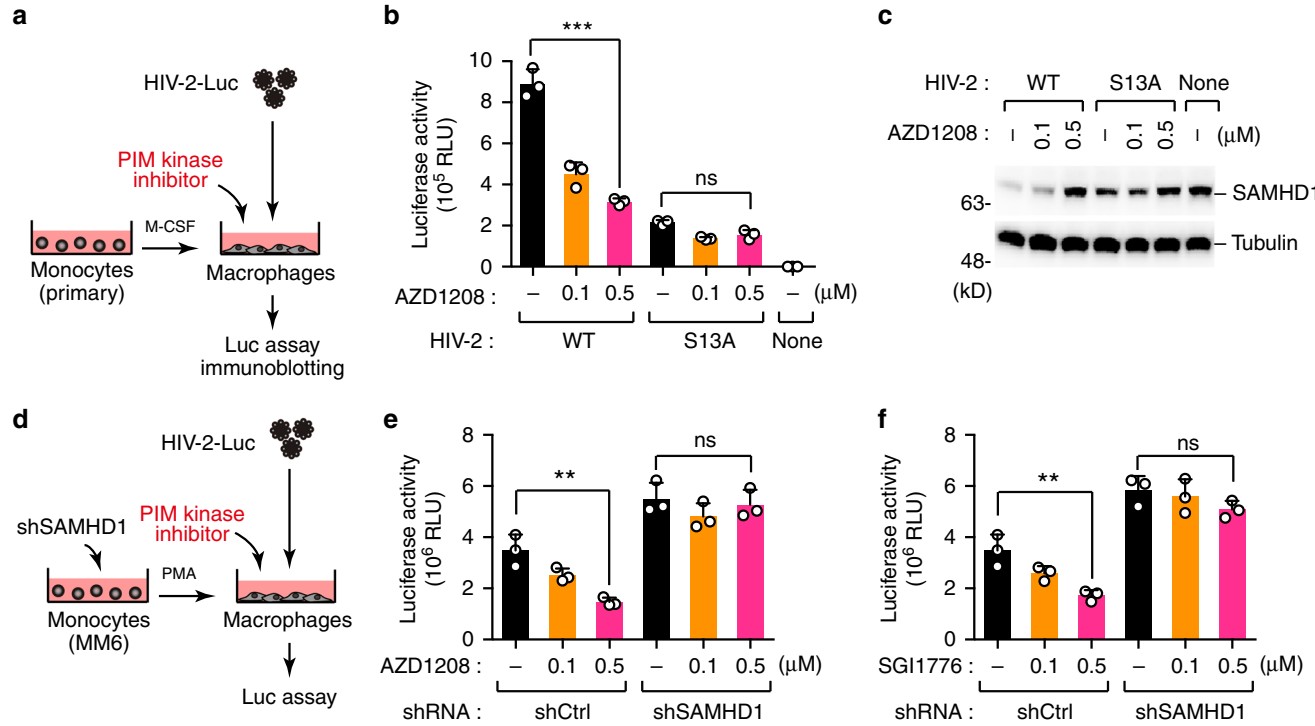

**Fig. 6** PIM kinase inhibition prevents Vpx-mediated SAMHD1 degradation. **a**–**c** PIM kinase inhibitors block HIV-2 infection in primary human monocyte–derived macrophages. Schematic representation of the experimental design (**a**). Primary monocytes were differentiated into macrophages and infected with HIV-2 reporter virus carrying wild-type Vpx or its S13A mutants. A pan-PIM kinase inhibitor AZD1208 was added 4 h before infection. Forty-eight hours post-infection, cells were harvested and subjected to luciferase assay (**b**) and immunoblotting (**c**) to measure HIV-2 infectivity and SAMHD1 expression, respectively. **d**–**f** The effect of PIM kinase inhibitors is SAMHD1-dependent. Schematic representation of the experimental procedure (**d**). Monomac6 (MM6) cells stably expressing shRNAs targeting SAMHD1 were differentiated into macrophages and infected with wild-type HIV-2 reporter virus. AZD1208 (**e**) or SGI1776 (**f**) was added 4 h before infection. Forty-eight hours post-infection, cells were harvested and subjected to luciferase assay to measure HIV-2 infectivity. All graphs are presented as a mean ± s.d. ($n = 3$). **$P < 0.01$; ***$P < 0.001$; ns, not significant, two-tailed unpaired $t$-test. Source data are provided as a Source Data file

lentiviral Vpx, we identified PIM family kinases (PIM1 and PIM3) as previously unrecognized positive regulators of lentiviral replication that enable viral evasion from SAMHD1-mediated restriction. Our current data indicate that these PIM family kinases target a highly conserved residue of Vpx, Ser13, and that this phosphorylation stabilizes the binding of Vpx to SAMHD1, resulting in efficient SAMHD1 degradation and viral replication (Fig. 7e). These findings demonstrate a previously undescribed paradigm in virus-host interaction ensuring the efficient lentiviral replication despite the existence of host restriction system.

The antiviral activity of SAMHD1 is limited to non-dividing cells such as macrophages and resting T cells, partially due to its role in limiting the supply of cytoplasmic dNTP. To overcome this potent antiviral factor, Vpx expropriates the host CRL4 E3 ubiquitin ligase complex and induces degradation of SAMHD1[5,6,9]. Although the mechanistic action of Vpx on SAMHD1 is well characterized, it is largely unknown how the antagonizing activity of Vpx is regulated. Previous studies have reported that host kinases FYN and ERK2 can phosphorylate tyrosine residues (Tyr66, 69, 71) of Vpx, thereby regulating its nuclear export and viral infectivity[14,15]. However, no previous study had reported evidence that Vpx phosphorylation is functionally relevant to viral escape from SAMHD1-mediated restriction. Our current results strongly suggest that the PIM family kinases PIM1 and PIM3 target a highly conserved residue of Vpx, Ser13, and that this phosphorylation affects the binding of Vpx to SAMHD1, and thus its activity. Moreover, we showed here that PIM kinase inhibitors selectively block lentiviral

replication by preventing Vpx-mediated SAMHD1 proteolysis. Notably, the pan-PIM kinase inhibitor AZD1208 has undergone clinical trials for hematological cancers[25]. Because PIM kinase promotes replication of Hepatitis C virus[26], AZD1208 might provide new therapeutic options against a series of virus infection.

PIM family kinases were originally identified as a target for proviral activation in T-cell lymphomas induced by murine leukemia virus[27]. Accordingly, PIM kinases are highly expressed in patients with lymphoma, leukemia, and prostate cancers[28,29]. Because elevated expression of PIM kinases prevents apoptosis, thereby increasing oncogenic activity, PIM kinases have been suggested as attractive drug targets in cancer. We found that AZD1208 has relatively high cytotoxicity in Monomac6-derived macrophages (Supplementary Fig. 11), presumably because this cell line was derived from human monoblastic leukemia, a known therapeutic target of AZD1208[20,30]. Importantly, however, this drug is much less toxic in primary macrophages, warranting its use in antiviral therapy. Although PIM family kinases are highly homologous to one another and share similar substrate specificity[31], our results showed that both PIM1 and PIM3 could phosphorylate Vpx Ser13. At present, we cannot exclude the possibility that PIM2 phosphorylates other residue(s) of Vpx. Additionally, our in vitro kinase assays revealed that PIM1 may more selectively phosphorylate Ser13, whereas PIM3 broadly phosphorylates other residues in Vpx as well (Figs. 1d and 2d). Detailed experiments should be needed to reveal the substrate specificity and functional significance of the Vpx phosphorylation in HIV-2/SIV replication.

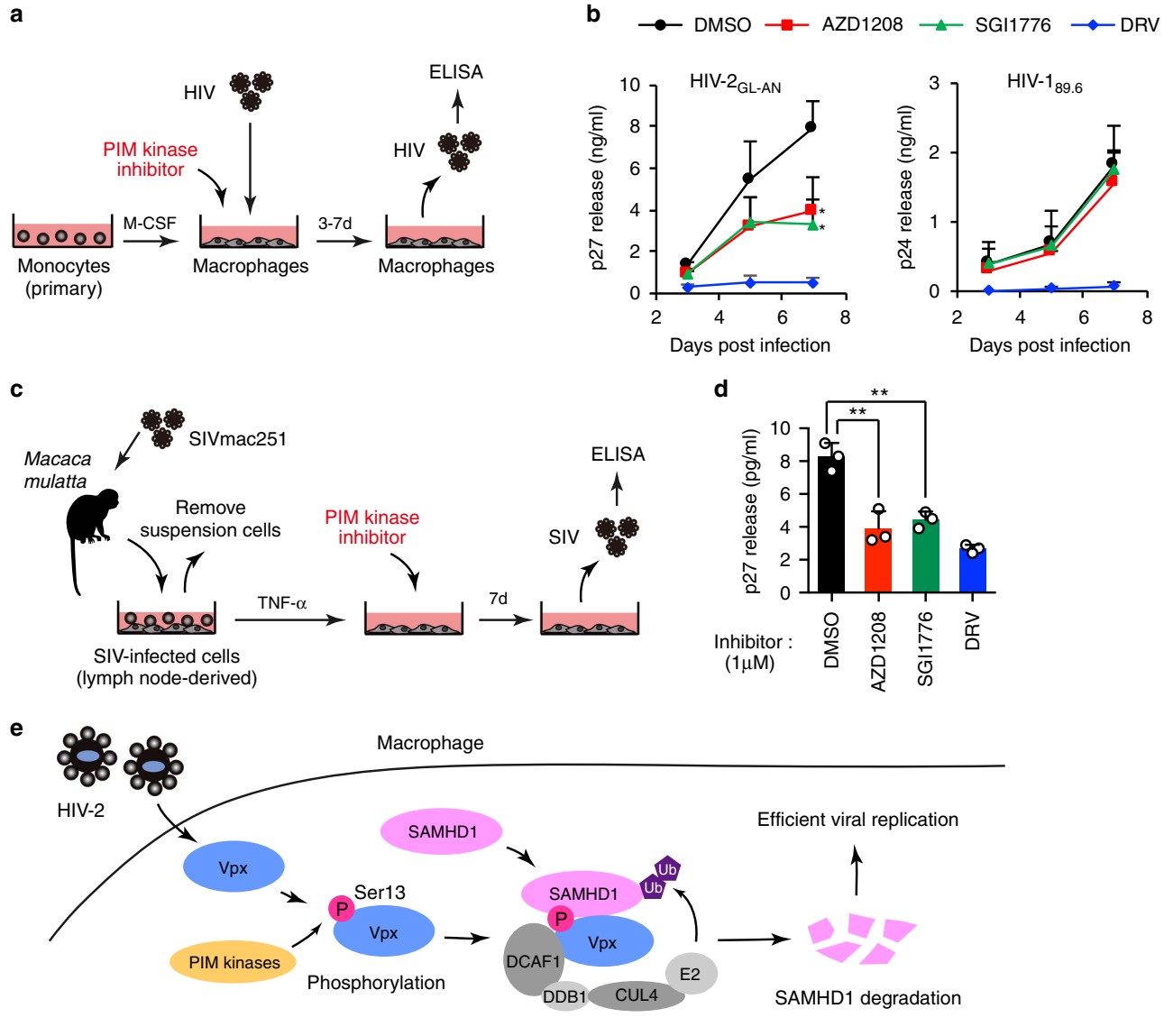

**Fig. 7** PIM kinase inhibitors block lentiviral replication in primary macrophages. **a**, **b** HIV-2 replication kinetics in primary monocyte–derived macrophages. Schematic representation of the experimental design (**a**). Cells were infected with wild-type HIV-2$_{GL-AN}$ or HIV-1$_{89.6}$ in the presence of AZD1208 or SGI1776 (1 μM). A HIV protease inhibitor darunavir (DRV, 1 μM) was used as a control. Replication kinetics were monitored by ELISA of extracellular viral p27 or p24 capsid antigens (**b**). **c**, **d** PIM kinase inhibitors can block SIV replication ex vivo. Schematic representation of the experimental procedure (**c**). Cells derived from a lymph node of an SIV-infected macaque monkey were seeded onto dishes. After suspension cells were removed, the remaining cells were stimulated with TNF-α (50 ng ml$^{-1}$) for 24 h, and then treated with either AZD1208, SGI1776, or DRV (1 μM) for 7 days. The levels of progeny virions in culture supernatants were determined by SIV p27 ELISA (**d**). **e** Schematic diagram illustrating the proposed role of PIM kinases in lentiviral infection. In infected macrophages, Vpx targets SAMHD1 for ubiquitination and proteasomal degradation by forming CRL4 E3 ubiquitin ligase complex. PIM family kinases (PIM1 and PIM3) target a highly conserved residue of Vpx, Ser13, and that this phosphorylation stabilizes the binding of Vpx to SAMHD1, resulting in efficient SAMHD1 degradation and viral replication. All graphs are presented as a mean ± s.d. ($n = 3$). *$P < 0.05$; **$P < 0.01$, two-tailed unpaired $t$-test. Source data are provided as a Source Data file

The amino acid residue Ser13 in Vpx is highly conserved among clinically isolated HIV-2 strains, whose Vpx proteins are capable of degrading SAMHD1. A mutagenesis study revealed that alanine substitution of Vpx Ser13 results in lower viral infectivity than the wild-type virus, even though the mutant protein is as stable as WT Vpx[32]. Our molecular dynamics simulation and subsequent biomolecular analyses clearly demonstrated that Vpx Ser13 phosphorylation could enhance SAMHD1 binding, presumably by forming an additional hydrogen bond between phospho-Vpx and SAMHD1. Consequently,

phosphorylation of Vpx enhances ubiquitin-mediated proteolysis of SAMHD1. Consistent with this, phosphorylation of the conserved amino acid of HIV proteins seems to be associated with viral escape from host antiviral immunity. For example, Vif-Ser144 and Vpu-Ser52,56 are highly conserved phospho-acceptor sites whose phosphorylation regulates antagonizing activity against the restriction factors APOBEC3G and Tetherin, respectively[33,34]. These post-translational modifications thus represent attractive targets for the development of novel anti-HIV agents that potentiate the host innate immune system.

## Methods

**AlphaScreen-based protein–protein interaction assays**. For the initial screen, we used a total of 412 complementary DNA library encoding human protein kinases[35,36]. DNA templates containing a biotin-ligating sequence were amplified by split-PCR with cDNAs and corresponding primers, and then used with the GenDecoder protein production system (CellFree Science)[37–39]. For synthesis of Vpx protein, *vpx* genes derived from the pGL-AN proviral plasmid[40] were generated by split-PCR and used as DNA templates. FLAG-tagged Vpx proteins were mixed with biotinylated kinases in 15 μl of reaction buffer (20 mM Tris–HCl pH 7.6, 5 mM MgCl₂, 1 mM DTT) in 384-well OptiPlates (PerkinElmer) and incubated at 26 °C for 1 h. Each sample was then mixed with AlphaScreen buffer containing anti-immunoglobulin G (protein A) acceptor beads and streptavidin-coated donor beads (0.1 μl each; PerkinElmer) and the anti-FLAG M2 antibody (5 μg ml⁻¹; Sigma-Aldrich), and further incubated at 26 °C. One hour later, AlphaScreen signals from the mixture were detected on an EnVision device (PerkinElmer) using the AlphaScreen signal detection program.

**NanoBRET-based protein–protein interaction assays**. Expression vectors encoding N-terminally HaloTag-conjugated host proteins (kinases, DCAF1 and SAMHD1) were prepared by Kazusa Genome Technologies (Chiba, Japan) and purchased from Promega. For NanoBRET analysis[41], HEK293 cells in white 96-well white plates were transfected with vectors encoding HaloTag-fused protein (100 ng) and NanoLuc-fused Vpx (1 ng). At 48 h post-transfection, NanoBRET activity was measured using the NanoBRET Nano-Glo Detection System (Promega).

**In vitro kinase assays**. Recombinant Vpx proteins were incubated with GST-tagged or His-tagged PIM kinases for 1 h at 37 °C in reaction buffer (20 mM Tris–HCl pH 7.5, 1 mM EDTA, 1 mM DTT, 150 mM NaCl, 5 mM MgCl₂, 0.05% Tween-20, 100 μM ATP, and 2 μCi [γ-³²P] ATP). The reaction mixture was subjected to electrophoresis on 10% SDS polyacrylamide gels, and the proteins were visualized on a BAS2500 image analyzer (Fujifilm, Japan). Alternatively, the proteins were subjected to immunoblotting using a phospho-specific polyclonal antibody against Vpx phosphorylated at Ser13 (produced by Scrum Inc., Japan).

**Liquid chromatography tandem-mass spectrometry analysis**. HEK293 cells expressing FLAG-Vpx and HA-PIM3, grown in 10-cm dishes, were immuno-precipitated with EZview Red FLAG M2 Affinity Gel (Sigma-Aldrich), and bound proteins were subjected to liquid chromatography tandem-mass spectrometry analysis. Bead-bound proteins were denatured with 8 M urea and 50 mM ammonium bicarbonate, and subsequently digested with trypsin for 16 h at 37 °C after reduction and alkylation. The resulting peptides were desalted using C18 Stage Tips[42] filled with C18 and SDB Empore disc membranes (3 M) and evaporated in a vacuum concentrator. Phosphopeptides were then enriched using Titansphere TiO₂ bulk beads (GL Sciences, Tokyo, Japan). After re-desalting with C18 stage Tips, phosphopeptides were analyzed on an LTQ Orbitrap Velos (Thermo Fisher Scientific) equipped with an UltiMate 3000 LC system (Thermo Fisher Scientific). Protein identification was performed using the MASCOT search engine, version 2.5.1 (Matrix Science) with the Swiss-Prot database (July 2014) with the following parameters: enzyme, trypsin; peptide mass tolerance, ± 5 ppm; fragment mass tolerance, ± 0.5 Da; maximum missed cleavage sites, 2; variable modifications: carbamidomethy-lation of cysteine, phosphorylation of serine or threonine, oxidation of methionine. We used a significance threshold of $p < 0.05$ as a cutoff to export results from the analysis by MASCOT. In addition, phosphopeptides that yielded a peptide ion score > 30 were considered positive identifications.

**Cells and virus preparation**. HEK293 cells (ATCC, #CRL-1573) were maintained in DMEM supplemented with 10% fetal bovine serum (FBS). THP-1 (JCRB, #0112) and U937 (JCRB, #9021) were cultured in RPMI (Wako) supplemented with 10% FBS. Monomac6 cells, a human monocyte-derived cell line exhibiting a mature monocyte phenotype[30], were kindly provided by Dr. Akinori Takaoka (Hokkaido University) and maintained in RPMI containing 10% FBS, 1 mM sodium pyruvate (Wako), 100 μg ml⁻¹ insulin (Sigma-Aldrich), and 1% non-essential amino acids (Thermo Fisher Scientific). For differentiation of U937 and Monomac6 cells to macrophages, the cells were differentiated for 24 h with 100 ng ml⁻¹ of phorbol 12-myristate 13-acetate (PMA, Sigma-Aldrich). Human CD14⁺ monocytes, purchased from PromoCell (#C-12909), were differentiated in RPMI supplemented with 10% FBS and 1000 U ml⁻¹ M-CSF (Wako) for 8 days. For transient knockdown, cells were transfected with gene-specific siRNA (Qiagen) using Lipofectamine RNAiMAX Transfection Reagent (Thermo Fisher Scientific). To generate PIM kinases- or SAMHD1-depleted Mono-mac6 cells, cells were transduced with lentiviral particles carrying gene-specific shRNA (Santa Cruz Biotechnology, catalog numbers #sc-36225-V, #sc-61353-V, and #sc-76442-V), and then selected with 1 μg ml⁻¹ puromycin (InvivoGen). HIV stocks were produced by transient transfection of HEK293 cells (10 cm dishes) with 10 μg of the molecular clone p89.6 (for HIV-1₈₉.₆)[43] or pGL-AN (for HIV-2_GL-AN)[40]. Vpx mutants were generated using PCR-based molecular cloning procedures. HIV-2-Luc reporter viruses were produced by co-transfection of HEK293 cells (10 cm dishes)

with 3 μg of VSV-G expression vector and either 10 μg of pGL-ANΔEnv-Luc or pGL-STΔEnv-Luc[18]. After 48 h, culture supernatants containing virus were collected and filtered through a 0.45-μm Millex-HV filter (Millipore).

**Immunoblotting and immunoprecipitation**. For immunoprecipitation analysis[44–46], HEK293 cells in six-well plates were transfected with vectors encoding FLAG-Vpx (500 ng) and HA-tagged protein expression plasmids (500 ng). At 48 h post-transfection, cells were lysed with HBST buffer (10 mM HEPES pH 7.4, 150 mM NaCl, 0.5% Triton-X-100) containing protease inhibitor Complete mini (Roche Diagnostics). Cell lysates were immunoprecipitated with anti-HA or FLAG EZview Red Affinity Gel (Sigma-Aldrich) for 16 h, and bound proteins were analyzed by immunoblotting as follows. Samples in SDS loading buffer were loaded onto 10% polyacrylamide gels, electrophoresed, and blotted onto PVDF membranes (Millipore). Membranes were probed with primary antibodies and horseradish peroxidase-conjugated secondary antibodies (GE Healthcare). Detected proteins were visualized using a FluorChem digital imaging system (Alpha Innotech). Band analysis was performed with ImageJ software (NIH). For SAMHD1 ubiquitination assays, HEK293 cells in six-well plates were transfected with vectors encoding FLAG-Vpx (500 ng), HA-SAMHD1 (1 μg), and Myc-ubiquitin (1 μg). Cells were treated with 2 μM MG132 (Sigma-Aldrich) for 18 h before being harvested. The antibodies used in this study are listed in Supplementary Table 1. Full images of all immunoblots are provided in Source Data file.

**Single-cycle HIV-2 infection assay**. Macrophages in 24-well plates were infected with VSV-G-pseudotyped HIV-2-Luc reporter viruses (2 ng of Gag p27 antigen). Forty-eight hours later, cells were subjected to immunoblot analysis and luciferase assay for calculating viral infectivity using Bright-Glo Luciferase Assay System (Promega). For experiments with PIM inhibitors, cells were treated with SGI1776 (Merck Millipore) or AZD1208 (MedChem Express) 4 h before infection.

**Multi-cycle viral replication assay**. Differentiated THP-1 cells in 24-well plates were infected with equal amounts of viruses (10⁵ reverse transcriptase units), and replication kinetics were calculated by reverse transcriptase activity of virus-containing culture supernatants[18]. For the experiment with PIM kinase inhibitors, macrophages in 24-well plates were infected with replication-competent HIV-2_GL-AN (5 ng of Gag p27 antigen) or HIV-1₈₉.₆ (20 ng of Gag p24 antigen) using the ViroMag Transfection Reagent (OZ Biosciences). Replication kinetics were calculated based on the level of extracellular viral capsid antigens using the SIV p27 or HIV p24 ELISA kit (ZeptoMetrix). For the experiment with SIV-infected cells, we used frozen cells derived from the lymph node of a macaque monkey infected with SIVmac251. The animal study was conducted under the guidelines provided by both Primate Research Institute (PRI) and Institute for Frontier Life and Medical Sciences (INFRONT), Kyoto University after the approval of Animal Welfare and Care Committee in the University. Research permission numbers were 2016–081 and 2017–116 for PRI and B15–3 and B15-3-2 for INFRONT, respectively. After removal of suspension cells, the remaining cells were stimulated with TNF-α (50 ng ml⁻¹) for 24 h, and then treated with either SGI1776, AZD1208, or darunavir (1 μM) for 7 days. The levels of progeny virions in the culture supernatants were determined using the SIV p27 ELISA kit (ZeptoMetrix).

**Molecular dynamics (MD) simulation**. The X-ray crystal structure of SAMHD1-Vpx-DCAF1 (Protein Data Bank code: 4CC9) was used as the template structure. On the basis of the predicted structural model of SAMHD1 with wild-type Vpx, three-dimensional structures of SAMHD1 with either unphosphorylated or phosphorylated Vpx were constructed using Molecular Builder in the Molecular Operating Environment software. MD simulations were performed by the pmemd module in the Amber 11 program package[47] with the AMBER ff99SB-ILDN force field[48] and the TIP3P water model for simulations of aqueous solutions[49]. A non-bonded cutoff of 10 Å was used. Bond lengths involving hydrogen were constrained with SHAKE, a constraint algorithm to satisfy Newtonian motion[50], and the time step for all MD simulations was set to 2 fs. After heating calculations for 20 ps until 310 K using the NVT ensemble, simulations were executed using the NPT ensemble at 1 atm, at 310 K, and in 150 mM NaCl for 100 ns.

**Statistical analysis**. All bar graphs present means and s.d. The statistical significance of differences between two groups was evaluated by two-tailed unpaired t-test in the Prism 6 software (GraphPad). A $p$ value > 0.05 was considered statistically significant.

## Data availability

The MS raw data have been deposited to the jPOSTrepo (Japan ProteOme STandard Repository) with the dataset identifier PXD013154. The source data underlying Figs. 1d, 2d–f, 3b-g, 4d-i, 5b–f, 6b–f, 7b and d, and Supplementary Figs. 4a, b, 5, 6a,b, 8a, b, 9, 10b, c and 11 are provided as a Source Data file.

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

## Acknowledgements

We thank Akiko Okayama for proteomic analysis, and Dr. Akinori Takaoka for providing Monomac6 cells. We also thank Noriko Saido, Mao Matsubayashi, and Masahito Matsumura for their technical assistance. The following reagents were obtained through the NIH AIDS Reagent Program, Division of AIDS, NIAID, NIH: HIV-2$_{ROD}$ Vpx Antiserum from Dr. Lee Ratner (Cat# 2609); Anti-HIV-1 p24 Monoclonal (183-H12-5C) from Dr. Bruce Chesebro and Kathy Wehrly (Cat# 3537); HIV-1$_{89.6}$ Infectious Molecular Clone (p89.6) from Ronald G. Collman (Cat# 3552). This work was supported by JSPS grants 16K08814 (to K.M.), and 16H05198 (to A.R.); by AMED grants JP18fk0410004, JP18fk0410020 (to K.M.) and JP18fk0410014 (to A.R.); by Yokohama Foundation for Advancement of Medical Science (to K.M.); and by the JST Creation of Innovation Centers for Advanced Interdisciplinary Research Areas Program (to A.R.).

## Author contributions

K.M. designed and performed the research, analyzed the data, and wrote the manuscript; S.M., M.Y., M.Nomaguchi and M.Nishi performed the research and analyzed the data; Y.K., H.K., H.S., H.H., T.T. and N.Y. analyzed the data; H.A., T.M. and A.A. contributed reagents and analyzed the data; T.S. developed the screening system; A.R. directed the research, analyzed the data, and wrote the manuscript.

## Additional information

**Competing interests:** The authors declare no competing interests.

