## [Peer Review File · Nature Communications]

Reviewers' comments:

Reviewer #1 (Remarks to the Author):

In this manuscript, Miyakawa et al. identified PIM kinases 1 and 3 as modulators of HIV infectivity by regulating Vpx S13 phosphorylation. The authors tend to show that this event is an important key in HIV infectivity since Vpx lacking this phosphorylation, close to its SAMHD1 binding site, seems to inhibit Vpx ability to counteract SAMHD1. This event also seems to strengthen interactions between the viral protein and this specific target so that Vpx efficiently ubiquitinylates this restriction factor leading to a well-known increase of infectivity. The authors also show that PIM kinases inhibitors are potent inhibitors of HIV-2 replication in primary cells, through SAMHD1 targeting, and they suggest that these drugs could be used as a novel therapeutic strategy.

This manuscript can be read with great fluidness, it's well written and clear. The little schemes showing how experiments were driven were very pleasant and helpful to understand the procedures leading to the presented results. Altogether, the manuscript brings to light a potentially interesting new mechanism controlling HIV-2 infectivity with experiments conducted in relevant primary cell models. However I would raise some issues that should be answered to strengthen the data and support the conclusions.

-The authors claimed that: 'An immunoprecipitation analysis further revealed that PIM1 and PIM3, but not PIM2, stably interact with Vpx' (Fig. 1C): Actually, PIM2 might interact with Vpx but with a lower affinity than Vpx with PIM1 and PIM3. Moreover, it seems the authors detect Vpx phosphorylation when recombinant PIM2 is added to recombinant Vpx (Fig. 1D) and this phosphorylation signal is stronger than with the use of recombinant FYN, previously reported to phosphorylate Vpx as the authors mentioned. Perhaps, PIM2 doesn't phosphorylate Vpx on S13, as demonstrated by the authors on Fig2.D, but on another residue (?).

In any event, the Co-IP experiment in Figure 1C lacks negative controls, that is to say an anti-Flag Immunoprecipitation in the presence of HA-PIM1 or HA-PIM3 but in the absence of Flag-Vpx, in case PIM1 or PIM3 would give some 'background' on the anti-Flag beads. Negative controls should be performed.

- Only HIV-2 viruses (Figures 3A to 3D for example) expressing luciferase have been used. It would be important to provide at least one experiment showing the impact of Vpx mutation (S13A) on the percentage of infected cells (with any kind of HIV-2 or HIV-1 virus), it could be done by intracellular Gag labelling or using a GFP virus; this would point to the SAMHD1-dependent pre-integration effect.

-I'm not totally confident with the assessment that HIV-2 S13A is less competent for SAMHD1 degradation, which is an important point at the basis of this study! SAMHD1 degradation by HIV-2 S13A seems really good in Figure 3G (and the little difference might result from different levels of Vpx expression?): the authors should at least show some quantification of the data (ratio SAMHD1/tubulin). In addition, would it be possible to show how Vpx S13A is expressed compared to Vpx wt?

-Again in Fig 3B, C, D: It would have been nice to show the SAMHD1 levels, by western blot, in these different macrophages.

-Negative controls are missing for the Co-IP Fig. 4F (with Flag-Vpx but without HA-SAMHD1). In addition, the numerical values should correspond to ratios of the immunoprecipitated material over the input. The authors should also immunoprecipitate Flag-Vpx first: this would have allowed to check that co-IP with DCAF1 was not affected by S13A, in contrast to co-IP with SAMHD1 and this would have validated panels D and E.

-Fig 4G: The SAMHD1(Ub)_n signal is not very strong. I wondered what would be the signal the authors could obtain with a Vpx-non-expressing condition. This lane with a total absence of signal (baseline) would reinforce that S13A mutant is not as efficient as WT to trigger SAMHD1 ubiquitinylation.

-Figures 5 B and C: I also recommend the authors to perform an infection with HIV-2-S13A, combined with the shPIM treatments. In these conditions, the fold decrease of luciferase expression should be reduced compared to this fold in WT conditions. This would reinforce the model.

-Figure 7: The effect of the drugs on HIV-2 replication is nice; however, we might wonder how far this supports the model. The authors should investigate the effect of the drugs on the replication of an HIV-1 virus that does not encode for Vpx (same as 7B but with HIV-1)

MINOR

-The title sounds too complicate (and the term "pivotal" is too strong).

-The authors nicely show (Figure 2) that Vpx, expressed in cells, is phosphorylated on Ser 13 using a specific phospho-serine antibody. Could the author provide the control experiment showing that transfected Flag-Vpx S13A along with with HA-PIM3 (same as 2E) is not recognized by the phospho-antibody?

-On page 8: 'However, phosphorylation of Vpx-Ser13 did create an additional hydrogen bond between Vpx and SAMHD1, leading to stabilization of the Vpx-SAMHD1 interaction (Fig. 4C).' I would use conditional here since the structural study (Fig 4B and C) is actually a simulation as mentioned by the authors. On the same page line 5, reference 19 does not deal with the structure of Vpx-DCAF1-SAMHD1.

-On page 9 line 8, 'Interestingly, when PIM kinases were depleted in producer cells, viral infectivity was not significantly affected (Fig. 5C)': It's not referring to Figure 5C here but 5E.

- The authors discuss about the toxicity of the PIM kinases inhibitors: 'We found that AZD1208 has relatively high cytotoxicity in Monomac6-derived macrophages [...]this drug is much less toxic in primary monocytes and macrophages, warranting its use in antiviral therapy.' Some test of toxicity should be provided, in particular when the drugs abrogate viral expression.

-The authors claim that 'the results indicate that PIM kinase-mediated phosphorylation of lentiviral Vpx is crucial for its function in down-regulation of SAMHD1'. Overall, they should moderate their statement.

Reviewer #2 (Remarks to the Author):

This is an interesting study investigating the molecular interactions between cellular protein kinases and HIV-2 Vpx. Vpx was previously shown to be phosphorylated by FYN kinase and the MAP kinase ERK-2 (Refs 14-15), which was associated with an effect on nuclear entry/export of Vpx and viral infectivity; however, the effect of Vpx phosphorylation on targeting SAMHD1 has not been explored in detail. To identify additional kinases involved in the phosphorylation of Vpx and their possible effect on targeting SAMHD1, the authors performed a comprehensive proteomics analysis of more than 400 human protein kinases to identify additional kinases involved in Vpx phosphorylation. In an AlphaScreen they identified the top 50 Vpx-interacting kinases, which were further narrowed down to three kinases using a cell-based NanoBret assay. They go on to identify PIM and PIM3 as the

kinases responsible for phosphorylation of Vpx at position S13 and they provide convincing data to support the conclusion that PIM-mediated phosphorylation of Vpx is critical for its ability to target SAMHD1. Overall, this is a carefully executed study that is interesting and timely. The manuscript is well-written, organized in a logical manner, and a pleasure to read. I do have a few comments for the authors' consideration.

Comments:

(1) In their secondary NanoBret assay (Fig. 1B), the authors identified Pim1, Pim3, as well as PDK1 as top Vpx-interacting kinases. They go on to investigate PIM1 and PIM3 but they all but forget about PDK1, which is not mentioned anywhere else in the manuscript. I don't expect the authors to include a comprehensive analysis of PDK1; but, can they at least explain their lack of enthusiasm for the PDK1 kinase?

(2) In Fig. 2D it appears that PIM1 and PIM3 equally affect phosphorylation of S13 in Vpx; In Fig. 1D, on the other hand, PIM3 appears to be clearly dominant. In Fig. 2D, a phosphoS13-specific antibody was used, while in Fig. 1D generic phosphorylation of Vpx was measured. One possible explanation would be that PIM1 selectively phosphorylates S13 while PIM3 phosphorylates other residues in Vpx as well. Can the authors comment on that and its possible functional implications?

(3) In Fig. 3, a S13A mutant was tested. I would find it interesting to include a phosphomimetic mutation (S13E) in this analysis, in analogy to similar experiments involving SAMHD1 (T592A/E).

(4) The authors make a strong case that phosphorylation of Vpx at S13 is important for targeting SAMHD1. However, the mechanistic consequences remain unclear. There is no predicted change in Vpx conformation (Fig. 4B) and the authors suggest that Vpx phosphorylation stabilizes its interaction with SAMHD1 (Fig. 4F). However, I wonder if Vpx phosphorylation simply stabilizes Vpx by reducing its proteasomal turnover. Evidence for that can be seen in Fig. 1C and 4F/G (although not in Fig. 2D, 2E).

(5) Fig. 3F: It would have been interesting to follow the experiment beyond 18 days since none of the infections had reached peak at the time the experiment was terminated.

(6) Fig. 3G: It is not clear, how long after infection the cells were analyzed for SAMHD1 expression. However, it is possible that not all cells in the S13 culture had been infected. It can therefore not be ruled out that Vpx-S13A retains full activity towards SAMHD1 depletion and that the SAMHD1 signal in Fig. 3G simply reflects residual uninfected cells. In that regard, the authors should state whether the cells used for the immunoblot were sorted for Gag expression or reflect bulk cultures. They should also include a Gag blot to support their statement concerning the reduced replication capacity of mutant viruses (p. 7).

Minor:

p. 9, line 9: Reference to Fig. 5C should read 5D (or D/E).

Reviewer #3 (Remarks to the Author):

The manuscript by Miyakawa et al. entitled "A pivotal role of PIM kinases in lentiviral evasion from the SAMHD1-mediated antiviral response" nicely describes the discovery of cellular kinases (PIM1 and PIM3 and to a lesser extent PIM2) that are capable of phosphorylating Vpx which leads to the enhancement of SAMHD1 degradation and thereby increased HIV-2 and SIVmac infection.

The manuscript is very interesting and it addresses an important aspect of the SAMHD1-Vpx field, how the degradation is regulated on a molecular level. The authors use state of the art screening methods and verify their findings by multiple complementing methods.

I very much support the publication of this manuscript, however the authors should address a couple of points before this, to even improve the key results that they are revealing.

1. In Fig1C PIM2 also seems to interact with Vpx, hence this is not a good negative control. In fact, the authors understate a possible role of PIM2 in their paper. While they acknowledge in agreement with published data that FYN phosphorylates Vpx (which I don't really see from the blot), they say that PIM2 does not (however the band appears even stronger in Fig.1D than for FYN.) Therefore I think the authors should tune down a negative role for PIM2 throughout the entire paper.
2. In Fig.1D a blot for Vpx is missing to compare the Vpx input. The phospho status is hard to compare without the total amount of Vpx being shown.
3. The authors should include size markers on their blots throughout the manuscript.
4. The authors show that S13A Vpx is similarly incorporated into particles in Fig.S4. This is an important control for Fig. 3B/C/D/F/G. However, in Fig.S4 it is done by 2 different assays: ELISA to determine p27 levels and WB to show Vpx levels. The authors should blot for p27 and show p27 levels by WB for the same gel they show Vpx levels for.

5. Fig.3F: error bars missing

6. Fig.4F: there seems to be also reduced levels of Vpx S13A in the input which could easily explain the differences seen in the IP...can this be quantified please.

7. Fig.5B S13A HIV2 should be used as a control here. PIM KD should have no effect on HIV-2 Vpx S13A. According to Fig.3B the RLU levels should be still in the 2000000nd so it should be possible to exclude a reduction for this virus when PIM3+1 are knocked down.

8. Fig.5D: PIM depletion should be shown in the 293T producer cells by WB.

9. Page 9: "AZD1208 'restored' SAMHD1 expression..." is not correct. It prevented SAMHD1 degradation.

10. Fig.7B needs error bars

11. Fig.7: The authors show effects on viruses that harbor a functional Vpx protein. I strongly suggest to also test an HIV-1 strain that replicates in MDMs (e.g. Bal) to show that this is indeed a Vpx effect of the drugs (HIV-1 lacks Vpx).

12. The authors suggest that phospho S13 leads to an additional hydrogen bonding with SAMHD1. The figure suggests that this occurs with SAMHD1 residue S616. Does Mutation of this residue lead to resistance for Vpx-induced degradation? The authors should discuss this possibility and possibly include some experimental data on a SAMHD1 S616 mutant, however this is not mandatory for publication of this manuscript. (would be a nice add on and would confirm their computer simulation of phospho-Vpx/SAMHD1 interaction)

Responses to the comments of Reviewer #1

“This manuscript can be read with great fluidness, it’s well written and clear. The little schemes showing how experiments were driven were very pleasant and helpful to understand the procedures leading to the presented results. Altogether, the manuscript brings to light a potentially interesting new mechanism controlling HIV-2 infectivity with experiments conducted in relevant primary cell models. However, I would raise some issues that should be answered to strengthen the data and support the conclusions.”

Response: We are grateful for the insightful and constructive suggestions by Reviewer #1. The reviewer noted some of the intriguing effects of PIM kinases on HIV-2 replication. However, he/she raised some issues that should be answered in order to strengthen our data and support our conclusions, in particular, the assessment that HIV-2 S13A is less capable of degrading SAMHD1. We have responded to these concerns in our answers below.

(#1-1) The authors claimed that: ‘An immunoprecipitation analysis further revealed that PIM1 and PIM3, but not PIM2, stably interact with Vpx’ (Fig. 1c): Actually, PIM2 might interact with Vpx but with a lower affinity than Vpx with PIM1 and PIM3. Moreover, it seems the authors detect Vpx phosphorylation when recombinant PIM2 is added to recombinant Vpx (Fig. 1d) and this phosphorylation signal is stronger than with the use of recombinant FYN, previously reported to phosphorylate Vpx as the authors mentioned. Perhaps, PIM2 doesn’t phosphorylate Vpx on S13, as demonstrated by the authors on Fig. 2d, but on another residue (?).

Response: We thank the reviewer for this insightful suggestion. As the reviewer pointed out, PIM2 may interact with Vpx with a lower affinity than PIM1 and PIM3, and seems to phosphorylate Vpx. However, it does not phosphorylate Ser13 of Vpx, which is the focus of our current study. To further investigate the functional role of PIM2 in HIV-2 replication, we performed a single-cycle infection assay in PIM2-depleted macrophages. Depletion of PIM2 did not affect virus replication (**Supplementary Fig. 10c**). Therefore, in this study, we focused on the effects on Vpx-Ser13 phosphorylation by PIM1/3. We will investigate the functional significance of PIM2-mediated phosphorylation of other residue(s) of Vpx in our future work. We have addressed these issues briefly in our revised text (**Page 15, line 299**).

(#1-2) In any event, the Co-IP experiment in Figure 1c lacks negative controls, that is to say an anti-Flag Immunoprecipitation in the presence of HA-PIM1 or HA-PIM3 but in the absence of Flag-Vpx, in case PIM1 or PIM3 would give some ‘background’ on the anti-Flag beads. Negative controls should be performed.

Response: We regret that we did not provide a negative control. Accordingly, we included anti-FLAG immunoprecipitation in the absence of FLAG-Vpx (**New Fig. 1c**).

(#1-3) Only HIV-2 viruses (Figures 3a to 3d for example) expressing luciferase have been used. It would be important to provide at least one experiment showing the impact of Vpx mutation (S13A) on the percentage of infected cells (with any kind of HIV-2 or HIV-1 virus), it could be done by intracellular Gag labelling or using a GFP virus; this would point to the SAMHD1-dependent pre-integration effect.

Response: We greatly appreciate this valuable suggestion. As requested, we stained intracellular Gag in monocytic cells infected with replication-competent HIV-2 (not reporter virus) and measured the percentage of infected cells by flow cytometry (**Supplementary Fig. 7a**). We confirmed that Vpx-S13A mutation decreased the percentage of infected cells while sustaining expression of SAMHD1. These data are now provided in **Supplementary Fig. 7b**. We also performed a multi-cycle infection assay with HIV-1 in the presence of a PIM kinase inhibitor (**New Fig. 7b**), as discussed below (#1-9).

(#1-4) I'm not totally confident with the assessment that HIV-2 S13A is less competent for SAMHD1 degradation, which is an important point at the basis of this study! SAMHD1 degradation by HIV-2 S13A seems really good in Figure 3g (and the little difference might result from different levels of Vpx expression?): the authors should at least show some quantification of the data (ratio SAMHD1/tubulin). In addition, would it be possible to show how Vpx S13A is expressed compared to Vpx wt?

Response: We completely agree. Indeed, Vpx-S13A can degrade SAMHD1, but less efficiently than wild-type. As requested, we added quantitative graphs showing that S13A is less able than wild-type Vpx to degrade SAMHD1 (**New Fig. 3b, 3d, 3g**).

We understand that the reviewer still has concerns that the above results are due to differences in the level of Vpx expression. To address this issue, we transfected FLAG-Vpx (wild-type or S13A) along with SAMHD1 expression plasmid because Vpx is very difficult to detect in HIV-2 infected cells. We found that Vpx-S13A is less able than wild-type to degrade SAMHD1, despite similar expression levels (**Supplementary Fig. 8a**). Moreover, a cycloheximide assay to examine the protein stability of wild-type and the S13A mutant revealed that both Vpx proteins have similar half-lives (**Supplementary Fig. 8b**). Together, these results indicate that S13 phosphorylation may be functionally important for SAMHD1 degradation.

(#1-5) Again in Fig 3b, c, d: It would have been nice to show the SAMHD1 levels, by western blot, in these different macrophages.

Response: As requested, we added immunoblots of SAMHD1 (**New Fig. 3b, 3c, 3d**).

(#1-6) Negative controls are missing for the Co-IP Fig. 4f (with Flag-Vpx but without HA-SAMHD1). In addition, the numerical values should correspond to ratios of the

immunoprecipitated material over the input. The authors should also immunoprecipitate Flag-Vpx first: this would have allowed to check that co-IP with DCAF1 was not affected by S13A, in contrast to co-IP with SAMHD1 and this would have validated panels d and e.

Response: We regret the lack of negative controls. To address this issue, we performed the co-IP assay with proper controls (**New Fig. 4h**). As suggested, we also performed the co-IP with FLAG-Vpx (**New Fig. 4f, 4g**). The results showed that Vpx-S13A weakened the interaction with SAMHD1, but not DCAF1.

(#1-7) Fig 4g: The SAMHD1(Ub)_n signal is not very strong. I wondered what would be the signal the authors could obtain with a Vpx-non-expressing condition. This lane with a total absence of signal (baseline) would reinforce that S13A mutant is not as efficient as WT to trigger SAMHD1 ubiquitinylation.

Response: As suggested, we added the Vpx–non-expressing condition as a negative control, and performed the experiment again. The new data are provided in **New Fig. 4i**. Although there is still some background from SAMHD1(Ub)_n signal in all lanes, the difference between wild-type and S13A is evident.

(#1-8) Figures 5b and c: I also recommend the authors to perform an infection with HIV-2-S13A, combined with the shPIM treatments. In these conditions, the fold decrease of luciferase expression should be reduced compared to this fold in WT conditions. This would reinforce the model.

Response: We thank the reviewer for pointing this out. As suggested, we compared the infectivity of wild-type and S13A virus in PIM1/3-depleted cells. As expected, we found that the infectivity of the S13A virus was comparable to that of the wild-type virus at a lower level in PIM-depleted cells (**New Fig. 5b, 5c**).

(#1-9) Figure 7: The effect of the drugs on HIV-2 replication is nice; however, we might wonder how far this supports the model. The authors should investigate the effect of the drugs on the replication of an HIV-1 virus that does not encode for Vpx (same as 7b but with HIV-1).

Response: We thank the reviewer for this important suggestion. As suggested, we investigated the effect of PIM inhibitors on HIV-1 replication in primary macrophages. We infected macrophages with a dual-tropic HIV-1 strain, 89.6, and then measured p24 levels over time. In accordance with our current model, these drugs had no observable effects on HIV-1 replication (**New Fig. 7b**).

MINOR

(#1-10) *The title sounds too complicate (and the term “pivotal” is too strong).*

Response: We agree. The title has been changed to **PIM kinases facilitate lentiviral evasion from the SAMHD1-mediated antiviral response via phosphorylation of Vpx.**”

(#1-11) *The authors nicely show (Figure 2) that Vpx, expressed in cells, is phosphorylated on Ser 13 using a specific phospho-serine antibody. Could the author provide the control experiment showing that transfected Flag-Vpx S13A along with HA-PIM3 (same as 2e) is not recognized by the phospho-antibody?*

Response: As suggested, we provided the western blotting data showing that Vpx-S13A is not recognized by the phospho-Ser13 antibody, with or without PIM3 expression (**Supplementary Fig. 4a**).

(#1-12) *On page 8: ‘However, phosphorylation of Vpx-Ser13 did create an additional hydrogen bond between Vpx and SAMHD1, leading to stabilization of the Vpx–SAMHD1 interaction (Fig. 4c).’ I would use conditional here since the structural study (Fig 4b and c) is actually a simulation as mentioned by the authors. On the same page line 5, reference 19 does not deal with the structure of Vpx-DCAF1-SAMHD1.*

Response: According to the reviewer’s suggestion, we toned down our description as follows: **“Our molecular simulation results suggested that phosphorylation of Vpx-Ser13 may create an additional hydrogen bond....” (page 9, line 178).** We also removed reference 19 from the revised manuscript.

(#1-13) *On page 9 line 8, ‘Interestingly, when PIM kinases were depleted in producer cells, viral infectivity was not significantly affected (Fig. 5c):’ It’s not referring to Figure 5c here but 5e.*

Response: We regret for the incorrect description, and have amended it in the revised manuscript.

(#1-14) *The authors discuss about the toxicity of the PIM kinases inhibitors: ‘We found that AZD1208 has relatively high toxicity in Monomac6-derived macrophages [...]this drug is much less toxic in primary monocytes and macrophages, warranting its use in antiviral therapy.’ Some test of toxicity should be provided, in particular when the drugs abrogate viral expression.*

Response: Accordingly, we performed a cytotoxicity assay, and added cell viability data obtained in the presence of the PIM kinase inhibitor AZD1208. This drug exerted a severe cytotoxic effect at concentrations above 4 μ M in Monomac6-derived macrophages, but no obvious cytotoxicity in primary monocyte-derived macrophages (**Supplementary Fig. 11**). Please note that we

performed several infection experiments at 0.1 and 1 μ M, concentrations at which there was no cytotoxic effect on Monomac6–derived macrophages (Figs. 6 and 7).

(#1-15) *The authors claim that ‘the results indicate that PIM kinase-mediated phosphorylation of lentiviral Vpx is crucial for its function in down-regulation of SAMHD1’. Overall, they should moderate their statement.*

Response: We appreciate this recommendation. We changed the text to “the results indicate that PIM kinase–mediated phosphorylation of lentiviral Vpx **regulates** its function in down-regulating SAMHD1” (Page 10, line 203).

Responses to the comments of Reviewer #2

“Overall, this is a carefully executed study that is interesting and timely. The manuscript is well-written, organized in a logical manner, and a pleasure to read. I do have a few comments for the authors’ consideration.”

Response: We sincerely appreciate the helpful and constructive evaluations by Reviewer #2. The reviewer regards our manuscript as interesting, but felt that additional experiments were required to support conclusions. Our responses to these concerns appear below.

(#2-1) *In their secondary NanoBret assay (Fig. 1b), the authors identified Pim1, Pim3, as well as PDK1 as top Vpx-interacting kinases. They go on to investigate PIM1 and PIM3 but they all but forget about PDK1, which is not mentioned anywhere else in the manuscript. I don’t expect the authors to include a comprehensive analysis of PDK1; but, can they at least explain their lack of enthusiasm for the PDK1 kinase?*

Response: We accept this as a valid concern. Indeed, we performed pilot studies showing that PDK1 does not phosphorylate Vpx Ser13, and that targeted depletion of PDK1 by shRNA does not affect HIV-2 replication. According to the reviewer’s suggestion, we provide these data as **Supplementary Fig. 4b and 10b**.

(#2-2) *In Fig. 2d it appears that PIM1 and PIM3 equally affect phosphorylation of S13 in Vpx; In Fig. 1d, on the other hand, PIM3 appears to be clearly dominant. In Fig. 2d, a phosphoS13-specific antibody was used, while in Fig. 1d generic phosphorylation of Vpx was measured. One possible explanation would be that PIM1 selectively phosphorylates S13 while PIM3 phosphorylates other residues in Vpx as well. Can the authors comment on that and its possible functional implications?*

Response: We appreciate this insightful comment. As the reviewer pointed out, our mass

spectrometry analysis revealed that PIM3 phosphorylated not only Ser13 but also Thr88 and Ser101 (Fig. 2a, Supplementary Fig. 3). Indeed, we also think that PIM3 is the predominant contributor to phosphorylation of Vpx Ser13, although PIM1 also phosphorylates this site, as revealed by immunoblotting using a phosphoS13-specific antibody. Unfortunately, we could not determine whether PIM1 can phosphorylate other sites in Vpx by mass spectrometry analysis, due to time and cost constraints. However, we think this issue is an important and should be addressed in our revised text (**Page 15, line 300**).

(#2-3) In Fig. 3, a S13A mutant was tested. I would find it interesting to include a phosphomimetic mutation (S13E) in this analysis, in analogy to similar experiments involving SAMHD1 (T592A/E).

Response: According to this suggestion, we made a reporter virus carrying Vpx-S13E and measured its infectivity. Our results indicated that both Vpx encapsidation and infectivity of the mutant virus were almost the same as those of wild-type viruses (**Supplementary Fig. 6a, 6b**). Also, we confirmed that this mutant virus had anti-SAMHD1 activity comparable to that of the wild-type virus in terms of SAMHD1 degradation (**Supplementary Fig. 6b, bottom**).

(#2-4) The authors make a strong case that phosphorylation of Vpx at S13 is important for targeting SAMHD1. However, the mechanistic consequences remain unclear. There is no predicted change in Vpx conformation (Fig. 4b) and the authors suggest that Vpx phosphorylation stabilizes its interaction with SAMHD1 (Fig. 4f). However, I wonder if Vpx phosphorylation simply stabilizes Vpx by reducing its proteasomal turnover. Evidence for that can be seen in Fig. 1c and 4f/g (although not in Fig. 2d, 2e).

Response: We completely agree that it is important to check the protein stability of Vpx wild-type and the S13A mutant. Therefore, we performed a cycloheximide assay, which revealed that S13A is as stable as wild-type (**Supplementary Fig. 8b**). To further address this concern, we performed immunoprecipitation assays again with appropriate negative controls and equal amounts of input (**New Fig 1c, 4f, 4h**). Together with the data about Vpx-S13E described above (#2-3), the results indicate that phosphorylation of Vpx at S13 is important for targeting SAMHD1.

(#2-5) Fig. 3f: It would have been interesting to follow the experiment beyond 18 days since none of the infections had reached peak at the time the experiment was terminated.

Response: As requested, we added data for 21 days post infection (**New Fig. 3f**). Indeed, HIV-2 infection had reached a peak by 18 days post infection.

(#2-6) Fig. 3g: It is not clear, how long after infection the cells were analyzed for SAMHD1 expression. However, it is possible that not all cells in the S13 culture had been infected. It can therefore not be ruled out that Vpx-S13A retains full activity towards SAMHD1 depletion and

that the SAMHD1 signal in Fig. 3g simply reflects residual uninfected cells. In that regard, the authors should state whether the cells used for the immunoblot were sorted for Gag expression or reflect bulk cultures. They should also include a Gag blot to support their statement concerning the reduced replication capacity of mutant viruses (p. 7).

Response: We regret the inadequate explanation of the time point. The experiment was performed at 12 days post infection; we now provided this information clearly in the legend of **New Fig. 3g**. Regarding the latter concern, we could not obtain enough samples to perform an immunoblot analysis. Instead, to analyze SAMHD1 expression in infected cells, we performed a flow cytometry analysis in which intracellular HIV-2 Gag-positive cells were gated (**Supplementary Fig. 7a**). Our results revealed that the S13A virus infected a lower percentage of cells (**Supplementary Fig. 7b**) and less potently decreased SAMHD1 expression than the wild-type virus (**Supplementary Fig. 7c**). These data indicate that Vpx-S13A does not retain full activity towards SAMHD1 depletion.

Minor:

(#2-7) p. 9, line 9: Reference to Fig. 5c should read 5d (or d/e).

Response: We have corrected this in the revised manuscript.

Responses to the comments of Reviewer #3

“The manuscript is very interesting and it addresses an important aspect of the SAMHD1-Vpx field, how the degradation is regulated on a molecular level. The authors use state of the art screening methods and verify their findings by multiple complementing methods.

I very much support the publication of this manuscript, however the authors should address a couple of points before this, to even improve the key results that they are revealing.”

Response: We greatly appreciate the careful analysis and constructive suggestions made by Reviewer #3. The reviewer suggested that we address a couple of important points before publication. We have revised the manuscript to address these concerns, as indicated below.

(#3-1) *In Fig1c PIM2 also seems to interact with Vpx, hence this is not a good negative control. In fact, the authors understate a possible role of PIM2 in their paper. While they acknowledge in agreement with published data that FYN phosphorylates Vpx (which I don't really see from the blot), they say that PIM2 does not (however the band appears even stronger in Fig.1d than for FYN.) Therefore, I think the authors should tune down a negative role for PIM2 throughout the entire paper.*

Response: We acknowledge that might have been misleading. As described above (#1-1), we agreed that PIM2 could phosphorylate Vpx on residue(s) other than Ser13. To further delineate the impact of PIM2 on HIV-2 replication, we performed an additional experiment showing that PIM2 depletion does not affect HIV-2 replication (**Supplementary Fig. 10c**). To avoid further confusion, we quantified the intensities of [³²P]-Vpx bands, showing that both FYN and PIM2 could phosphorylate Vpx (**New Fig. 1d**). We will investigate the functional significance of PIM2-mediated phosphorylation of other residue(s) of Vpx in our future work. We have addressed these issues briefly in our revised text (**Page 15, line 299**).

(#3-2) *In Fig.1d a blot for Vpx is missing to compare the Vpx input. The phospho status is hard to compare without the total amount of Vpx being shown.*

Response: We regret the omission. We added a Vpx blot in **New Fig. 1d**.

(#3-3) *The authors should include size markers on their blots throughout the manuscript.*

Response: We appreciate the suggestion. We included size markers on all immunoblots throughout the manuscript.

(#3-4) *The authors show that S13A Vpx is similarly incorporated into particles in Fig.S4. This is an important control for Fig. 3b/c/d/f/g. However, in Fig.S4 it is done by 2 different assays: ELISA to determine p27 levels and WB to show Vpx levels. The authors should blot for p27 and show p27 levels by WB for the same gel they show Vpx levels for.*

Response: We appreciate this valuable comment. We added the p27 blot in **Supplementary Fig. 5**.

(#3-5) *Fig.3f: error bars missing.*

Response: We added error bars accordingly (**New Fig. 3f**).

(#3-6) *Fig.4f: there seems to be also reduced levels of Vpx S13A in the input which could easily explain the differences seen in the IP...can this be quantified please.*

Response: We agree with the reviewer. We repeated the IP analysis with appropriate negative controls and equal levels of input, and have added quantitations of the precipitants to **New Fig. 4f-h**. According Reviewer #1's suggestion (#1-6), we added data from immunoprecipitations with anti-FLAG antibody (**New Fig. 4f, 4g**).

(#3-7) *Fig.5b S13A HIV2 should be used as a control here. PIM KD should have no effect on HIV-2 Vpx S13A. According to Fig.3B the RLU levels should be still in the 2000000nd so it should be possible to exclude a reduction for this virus when PIM3+1 are knocked down.*

Response: According to this suggestion, we repeated the experiments with wild-type and S13A HIV-2. As expected, we found that the infectivity of the Vpx-S13A virus was comparable to that of the wild-type virus in PIM1/3-depleted macrophages. This data has been added in **New Fig. 5b, 5c.**

(#3-8) *Fig.5d: PIM depletion should be shown in the 293T producer cells by WB.*

Response: We added a blot of PIM1 and PIM3 in 293T cells transduced with siRNAs (**New Fig. 5f**).

(#3-9) *Page 9: “AZD1208 ‘restored’ SAMHD1 expression...” is not correct. It prevented SAMHD1 degradation.*

Response: This has been corrected (**Page 10, line 201; Page11, line 220**).

(#3-10) *Fig.7b needs error bars.*

Response: We added error bars to **New Fig. 7b**.

(#3-11) *Fig.7: The authors show effects on viruses that harbor a functional Vpx protein. I strongly suggest to also test an HIV-1 strain that replicates in MDMs (e.g. Bal) to show that this is indeed a Vpx effect of the drugs (HIV-1 lacks Vpx).*

Response: As discussed above (#1-9), we investigated the effect of PIM inhibitors on HIV-1 replication. We infected primary macrophages with a dual-tropic HIV-1 strain, 89.6, and then measured p24 levels over time. Notably, these drugs did not affect HIV-1 replication. These data are provided in **New Fig. 7b**.

(#3-12) *The authors suggest that phospho S13 leads to an additional hydrogen bonding with SAMHD1. The figure suggests that this occurs with SAMHD1 residue S616. Does Mutation of this residue lead to resistance for Vpx-induced degradation? The authors should discuss this possibility and possibly include some experimental data on a SAMHD1 S616 mutant, however this is not mandatory for publication of this manuscript. (would be a nice add on and would confirm their computer simulation of phospho-Vpx/SAMHD1 interaction)*

Response: We appreciate these suggestions. To address these concerns, we created the SAMHD1-S616A mutant and assessed its degradation by Vpx. As described in **Supplementary Fig. 9**, the SAMHD1-S616A mutation gives rise to resistance against Vpx-induced degradation. This data suggests that SAMHD1 Ser616 is an important residue for the Vpx–SAMHD1 interaction.

REVIEWERS' COMMENTS:

Reviewer #1 (Remarks to the Author):

This is a very nice body of work that brings together Vpx phosphorylation by PIM kinases, Vpx ability to induce SAMHD1 degradation and HIV-2 infectivity. Altogether the authors nicely show how PIM kinase-mediated phosphorylation of Vpx contributes to SAMHD1 degradation and HIV-2 infectivity.

The authors have clearly answered my concerns. In particular, they performed well-controlled and beautiful co-IP experiments; they also now show the lack of effect of PIM kinase depletion on a HIV-2-S13A virus and the lack of effect of PIM kinase inhibitors on HIV-1 that does not encode for Vpx.

Altogether, I recommend publication. The following comments are very minor and can be answered easily without a new revision.

-New Fig. 3b, 3d, 3g: When SAMHD1 quantification is performed, ratios of SAMHD1 over tubulin should be given (and not only SAMHD1 quantification)

-Supplementary Fig. 8b is not very convincing. Nonetheless, I really do appreciate the new blots Fig. 3b-d and the new co-IP Figures 4f-h, which clearly show the differences between wt Vpx and S13A mutant towards SAMHD1.

-I would suggest an additional change in the title: "PIM kinases facilitate lentiviral evasion from SAMHD1 restriction via Vpx phosphorylation".

Florence Margottin-Goguet

Reviewer #2 (Remarks to the Author):

The authors have addressed my previous critiques to my full satisfaction. I have no further suggestions for improvement

Reviewer #3 (Remarks to the Author):

The authors have done a great job to address all my concerns and requests for additional data and controls. The HIV-1 data (Fig.7b) is an important control demonstrating specificity for HIV-2. In addition, the KD of PIM1 and 3 only affects wild type but not the Vpx S13A mutant (Fig.5b), demonstrating specificity towards Vpx residue S13. To me these are the most important experiments demonstrating that PIM1 and 3 affect HIV-2 replication in a Vpx-dependent manner. It will be interesting for future studies to address the redundancy between the different PIM kinases. I recommend the publication of this manuscript in Nature Communications and would like to thank the authors for this interesting and sound study. Torsten Schaller

Responses to the comments of Reviewer #1

“This is a very nice body of work that brings together Vpx phosphorylation by PIM kinases, Vpx ability to induce SAMHD1 degradation and HIV-2 infectivity. Altogether the authors nicely show how PIM kinase-mediated phosphorylation of Vpx contributes to SAMHD1 degradation and HIV-2 infectivity.

The authors have clearly answered my concerns. In particular, they performed well-controlled and beautiful co-IP experiments; they also now show the lack of effect of PIM kinase depletion on a HIV-2-S13A virus and the lack of effect of PIM kinase inhibitors on HIV-1 that does not encode for Vpx.

Altogether, I recommend publication. The following comments are very minor and can be answered easily without a new revision.”

Response: We wish to express our highest appreciation to the Reviewer#1 for he/her insightful suggestions, which have helped us significantly improve the paper. We have addressed the comments of this reviewer as follows:

(#1-1) New Fig. 3b, 3d, 3g: When SAMHD1 quantification is performed, ratios of SAMHD1 over tubulin should be given (and not only SAMHD1 quantification).

Response: Accordingly, we have provided the bar charts showing the ratio of SAMHD1 to tubulin in **New Figure 3b, 3d, and 3g**.

(#1-2) Supplementary Fig. 8b is not very convincing. Nonetheless, I really do appreciate the new blots Fig. 3b-d and the new co-IP Figures 4f-h, which clearly show the differences between wt Vpx and S13A mutant towards SAMHD1.

Response: Although the reviewer has still some concerns with our protein stability data, we would appreciate that he/she has basically accepted our conclusion that S13A is less competent in SAMHD1 interaction/degradation rather than its protein stability. Indeed, as the reviewer pointed out, our comprehensive analysis supported the conclusion (Fig. 3b-d, 3g, 4f-h, 5b, 6b, and Supplementary Fig. 8a).

(#1-3) I would suggest an additional change in the title: “PIM kinases facilitate lentiviral evasion from SAMHD1 restriction via Vpx phosphorylation”.

Response: We acknowledge the valid suggestion. We have changed the title accordingly.

Responses to the comments of Reviewer #2

“The authors have addressed my previous critiques to my full satisfaction. I have no further suggestions for improvement.”

Response: We again deeply appreciate the careful analysis and constructive suggestions made by Reviewer#2.

Responses to the comments of Reviewer #3

“The authors have done a great job to address all my concerns and requests for additional data and controls. The HIV-1 data (Fig.7b) is an important control demonstrating specificity for HIV-2. In addition, the KD of PIM1 and 3 only affects wild type but not the Vpx S13A mutant (Fig.5b), demonstrating specificity towards Vpx residue S13. To me these are the most important experiments demonstrating that PIM1 and 3 affect HIV-2 replication in a Vpx-dependent manner. It will be interesting for future studies to address the redundancy between the different PIM kinases. I recommend the publication of this manuscript in Nature Communications and would like to thank the authors for this interesting and sound study.”

Response: We sincerely appreciate the helpful and constructive suggestions by Reviewer#3. We believe that our manuscript has been much improved by these suggestions.